# BlendRL: A Framework for Merging Symbolic and Neural Policy Learning

**Hikaru Shindo**[1]  **Quentin Delfosse**[1]  **Devendra Singh Dhami**[2]  **Kristian Kersting**[1,3,4,5]

[1]Department of Computer Science, Technical University of Darmstadt, Germany
[2]Dept. of Mathematics and Computer Science, Eindhoven University of Technology, Netherlands
[3]Hessian Center for Artificial Intelligence (hessian.AI), Germany
[4]German Research Center for Artificial Intelligence (DFKI), Germany
[5]Centre for Cognitive Science, Technical University of Darmstadt, Germany
{hikaru.shindo, quentin.delfosse, kersting}@tu-darmstadt.de, d.s.dhami@tue.nl

## Abstract

Humans can leverage both abstract reasoning and intuitive reactions. In contrast, reinforcement learning policies are typically encoded in either opaque systems like neural networks or symbolic systems that rely on predefined symbols and rules. This disjointed approach severely limits the agents' capabilities, as they often lack either the flexible low-level reaction characteristic of neural agents or the interpretable reasoning of symbolic agents. To overcome this challenge, we introduce *BlendRL*, a neuro-symbolic RL framework that harmoniously integrates both paradigms within RL agents that use mixtures of both logic and neural policies. We empirically demonstrate that BlendRL agents outperform both neural and symbolic baselines in standard Atari environments, and showcase their robustness to environmental changes. Additionally, we analyze the interaction between neural and symbolic policies, illustrating how their hybrid use helps agents overcome each other's limitations.

## 1 Introduction

To solve complex problems, humans employ two fundamental modes of thinking: (1) instinctive responses for immediate reaction and motor control and (2) abstract reasoning, using distinct identifiable concepts. These two facets of human intelligence are commonly referred to as *System 1* and *System 2* (Kahneman, 2011). While our reasoning system requires symbols to build interpretable decision rules, instinctive reactions do not rely on such inductive bias, but lack transparency. Both systems are required in challenging RL environments, such as *Kangaroo*, where the agent's goal is to reach its joey (at the top), captured by monkeys that need to be punched out of the way, or in *Seaquest*, where the agent controls a submarine, and needs to collect swimming divers without running out of oxygen (*cf.* Figure 1). Developing agents that can effectively use both information processing systems has proven to be a persistent challenge (Lake et al., 2017; Mao et al., 2019; Kautz, 2022). The main difficulty not only lies in achieving high-level capabilities with both systems, but also in seamlessly integrating these systems to interact synergistically, in order to maximize performances without sacrificing transparency.

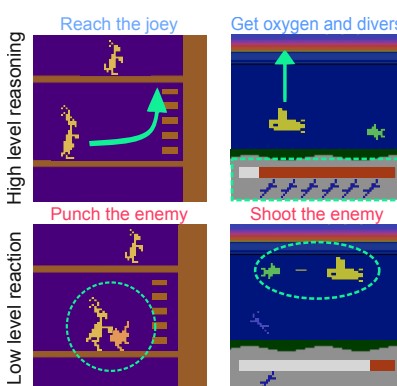

Figure 1: **Most problems require both *reasoning* (top) and *reacting* (bottom)**, to master *e.g.* the Kangaroo (left) and Seaquest (right) Atari games.

Deep neural networks have demonstrated an ability to effectively learn policies across a wide range of tasks without relying on any prior knowledge about the task (Mnih et al., 2015; Schulman et al., 2017; Badia et al., 2020; Bhatt et al., 2024). However, these black box policies can leverage shortcuts that are imperceptible to human observers (Locatello et al., 2020; Liu & Borisyuk, 2024). For instance, in the simple Atari Pong game, deep agents tend to rely on the enemy's position rather than the ball's one (Delfosse et al., 2024d), demonstrating deep learning systems' tendency to rely on shortcut learning opportunities, that then fail to generalize to slightly modified environments.

To enhance reasoning capabilities, symbolic reasoning has been integrated into approaches, using *e.g.* as logic-based policies (Jiang & Luo, 2019; Kimura et al., 2021; Cao et al., 2022; Delfosse et al., 2023a) or program-based frameworks (Sun et al., 2020; Verma et al., 2018; Lyu et al., 2019; Cappart et al., 2021; Kohler et al., 2024). These methods offer transparency, revisability, better generalization, and the potential for curriculum learning. However, they often rely on the incorporation of specific human inductive biases, necessary for solving the tasks, thus requiring experts to provide essential concepts or potential logic rules. Moreover, actions involving low-level reactions with subtle movements are challenging, if not impossible, to encode within these frameworks. This limitation underscores the constraints of symbolic systems in terms of their learning capabilities. Consequently, an important question emerges: *How can we build intelligent agents that leverage the strengths of both neural and symbolic modeling?* The current approach to combining these systems typically uses a top-down (*i.e.* sequential) method: using deliberative systems (*e.g.* planners) to provide high-level reasoning to select reactive systems (*e.g.* Deep RL), that offer quick, low-level responses (Kokel et al., 2021). However, this top-down approach is not always suitable, *e.g.* , a self-driving can recompute its plan on a low-density highway, but must react quickly in heavy traffic, without replanning. Agents that can select for either neural or symbolic modeling based on the context are necessary.

We introduce *BlendRL*, a framework that integrates neural and logic-based policy learning in parallel. BlendRL agents learn logic-based interpretableee reasoning as well as low-level control, combined through a blending function that utilizes a hybrid state representation. They can utilize high-level reasoning (*e.g.* for path finding), which benefits from a symbolic (or object-centric) state representation, and low-level reactions, for finetuned control skills (*e.g.* shooting at enemies), using pixel-based state representations. Although its neural component is less interpretable, it assists the agent in adapting to situations where it lacks sufficient symbolic representations. BlendRL provides hybrid policies, by distinctly modeling these two types of information processing systems, and selects its actions using a mixture of deep neural networks and differentiable logic reasoners (Evans & Grefenstette, 2018; Shindo et al., 2023; 2024b). Additionally, we propose an Advantage Actor-Critic (A2C)-based learning algorithm for BlendRL agents, that incorporates Proximal Policy Optimization (PPO) and policy regularization, supplemented by analysis on the interactions between neural and symbolic components of trained agents. Overall, we make the following contributions:

**(i)** We propose BlendRL to *jointly and simultaneously* train symbolic and neural policies.

**(ii)** For efficient learning on the proposed framework, we adapt the PPO Actor Critic algorithm on the hybrid state representation. Moreover, we proposed a regularization method to balance neural and symbolic policies, providing agents that are both transparent reasoners and accurate reactors.

**(iii)** We empirically show that BlendRL agents outperform neural and the state-of-the-art neuro-symbolic baselines on environments where agents need to perform both high-level reasoning and low-level reacting. Moreover, we demonstrate the robustness of the BlendRL agents to environmental changes.

**(iv)** We provide a deeper analysis of the interactions between neural and symbolic policies, revealing how hybrid representations and policies can help agents overcome each other's limitations.

We start off by providing the necessary background, then introduce our BlendRL method for policy reasoning and learning. We experimentally evaluate BlendRL on three complex Atari games, comparing its performance to purely neural and logic baselines. Following this, we discuss related work before concluding. Our code and resources are openly available.[1]

---

[1] https://github.com/ml-research/blendrl

## 2 BACKGROUND

Let us introduce the necessary background before formally introducing our BlendRL method.

**Deep Reinforcement Learning**. In RL, the task is modelled as a Markov decision process, $\mathcal{M} =<\mathcal{S}, \mathcal{A}, P, R, \gamma>$, where, at every timestep $t$, an agent in a state $s_t \in \mathcal{S}$, takes action $a_t \in \mathcal{A}$, receives a reward $r_t = R(s_t, a_t)$ and a transition to the next state $s_{t+1}$, according to environment dynamics $P(s_{t+1}|s_t, a_t)$. Deep agents attempt to learn a parametric policy, $\pi_\theta(a_t|s_t)$, in order to maximize the return (*i.e.* $\sum_t \gamma^t r_t$, with $\gamma \in [0, 1]$). The desired input to output (*i.e.* state to action) distribution is not directly accessible, as RL agents only observe returns. The value $V_{\pi_\theta}(s_t)$ (resp. Q-value $Q_{\pi_\theta}(s_t, a_t)$) function provides the return of the state (resp. state/action pair) following the policy $\pi_\theta$. We provide more details in App. A.13.

**First-Order Logic (FOL).** In FOL, a *Language* $\mathcal{L}$ is a tuple $(\mathcal{P}, \mathcal{D}, \mathcal{F}, \mathcal{V})$, where $\mathcal{P}$ is a set of predicates, $\mathcal{D}$ a set of constants, $\mathcal{F}$ a set of function symbols (functors), and $\mathcal{V}$ a set of variables. A *term* is either a constant (*e.g.* `obj1`, `agent`), a variable (*e.g.* `O1`), or a term which consists of a function symbol[2]. An *atom* is a formula $p(t_1, \ldots, t_n)$, where p is a predicate symbol (*e.g.* `closeby`) and $t_1, \ldots, t_n$ are terms. A *ground atom* or simply a *fact* is an atom with no variables (*e.g.* `closeby(obj1, obj2)`). A *literal* is an atom ($A$) or its negation ($\neg A$). A *clause* is a finite disjunction ($\vee$) of literals. A *ground clause* is a clause with no variables. A *definite clause* is a clause with exactly one positive literal. If $A, B_1, \ldots, B_n$ are atoms, then $A \vee \neg B_1 \vee \ldots \vee \neg B_n$ is a definite clause. We write definite clauses in the form of $A :\!\!- B_1, \ldots, B_n$. $A$ is the rule *head*, and the set $\{B_1, \ldots, B_n\}$ is its *body*. We interchangeably use definite clauses and *rules*.

**Differentiable Forward Reasoning** is a data-driven approach to reasoning in First-Order Logic (FOL) (Russell & Norvig, 2010). In forward reasoning, given a set of facts and rules, new facts are deduced by applying the rules to the facts. Differentiable forward reasoning is a differentiable implementation of forward reasoning, utilizing tensor-based differentiable operations (Evans & Grefenstette, 2018; Shindo et al., 2023) or graph-based approaches (Shindo et al., 2024b). This approach can be efficiently applied to reinforcement learning tasks by encoding actions in the form of rules, where the head defines an action and the body specifies its conditions. To learn the *importance* or truth value of each rule, they can be associated with learnable rule weights. Consequently, hypotheses can be formulated in terms of rules and learned from data.

## 3 BLENDRL

BlendRL integrates both abstract reasoning and instinctive reactions by combining symbolic and neural policies. As illustrated in Figure 2, the neural policy processes sub-symbolic (*i.e.* pixel-based) representations to compute a distribution over the actions, while the reasoning module employs differentiable reasoning on symbolic states. These action distributions are then blended to obtain the final action distribution. Let us point out that the logic reasoner does not use planning. BlendRL produces model free RL agents. We begin by describing the inner workings of each policy type, and of the blending module. Next, we discuss how we leverage the common sense encapsulated in pretrained large language models (LLMs) to obtain symbolic concepts and their evaluation functions. Finally, we describe how we adapted the PPO actor-critic algorithm to perform end-to-end training of BlendRL modules. Let us first formally introduce the state representations.

### 3.1 HYBRID STATE REPRESENTATIONS

BlendRL agents employ *two* distinct state representations: (i) *pixel-based representations* and (ii) *object-centric representations*, that can be extracted by object discovery models (Redmon et al., 2016; Lin et al., 2020; Delfosse et al., 2023b; Zhao et al., 2023). Pixel-based representations usually consist of a stack of

---

[2]In our experiments, we opted to use rules without function symbols as they were not necessary (*cf.* Fig 4). However, BlendRL agents are capable of handling them. (*cf.* App. A.1.3)

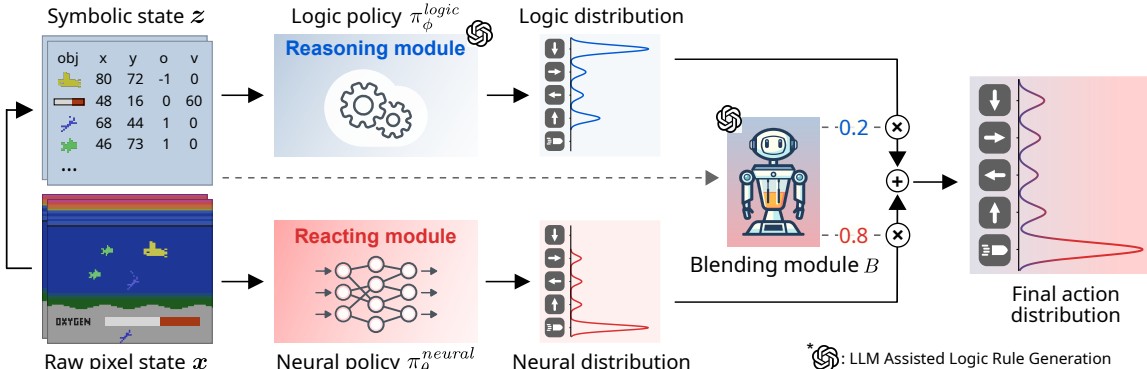

Figure 2: **Overview of the BlendRL framework**. BlendRL employs two policy types to compute action probabilities. A deep neural policy (bottom) handles low-level reactions, operating on pixel-based states, while differentiable forward reasoners (Shindo et al., 2023) (top) manage high-level reasoning on symbolic, object-centric states extracted from the pixel-based input. The blending module then merges the two action distributions by taking a weighted sum based on the current state. The logic policy and the blending module incorporate human inductive biases extracted using language models (LLMs) and the task context. All three components (the two policies and the blending module) can be trained jointly using gradients.

raw images, provided by the environment, fed to a deep convolutional network, as introduced by Mnih et al. (2015). Our considered symbolic (object-centric) representations consist of a list of objects, with attributes (*e.g.* position, orientation, color, etc.), allowing for logical reasoning on structured representations (Zada-ianchuk et al., 2021; Liu et al., 2021; Yoon et al., 2023; Wüst et al., 2024; Stammer et al., 2024).

Raw sub-symbolic states, $x \in \mathbb{R}^{F \times W \times H \times C}$, consist of the last $F$ observed frames, while symbolic states $z \in \mathbb{R}^{n \times m}$ represents the corresponding symbolic (or object-centric) state, with $n$ denoting the number of objects and of $m$ extracted properties. As illustrated in Figure 2, on the *Seaquest* Atari environment, a symbolic state consists of objects such as the agent's submarine, the oxygen level, a diver, sharks, etc., with their $x$ and $y$ positional coordinates, as well as orientation and value. If a property is not relevant to an object (*e.g.* the orientation for the oxygen bar), it is set to 0.

## 3.2 THE BLENDED NEURO-SYMBOLIC POLICIES

Using both state representations, BlendRL agents compute the action selection probabilities by aggregating the probabilities of its *neural* and its *logic* policies.

**The neural policy**, $\pi_\theta^{neural} : \mathbb{R}^{F \times W \times H \times C} \rightarrow [0, 1]^A$, consists of a neural network, parameterized by $\theta$, that computes action distributions based on pixel state representations ($x$). Convolutional neural networks based policies (Mnih et al., 2015; Schulman et al., 2017; Hessel et al., 2018) are the most common deep policy types (Huang et al., 2022), but visual transformers can also be used (Chen et al., 2021; Parisotto et al., 2020).

**The logic policy**, $\pi_\phi^{logic} : \mathbb{R}^{n \times m} \rightarrow [0, 1]^A$, is a differentiable forward reasoner (Shindo et al., 2023), parameterized by $\phi$, that reasons over object-centric states, such as the one depicted in Figure 2. Action rules are rules in first-order logic that encode conditions for an action to be selected (Reiter, 2001; Delfosse et al., 2023a). An action rule defines an action as its head atom (*action atom*) and encodes its preconditions in its body atoms (*state atoms*). We provide exemplary action rules for *Seaquest* in Listing 1.

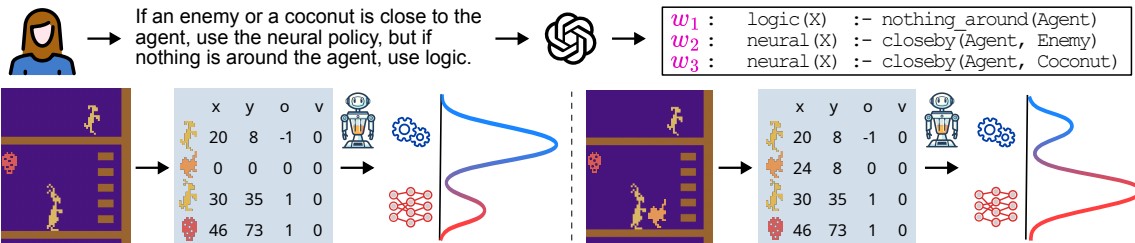

Figure 3: **LLMs allow for generating transparent blending function**, **Blending** exemplified on *Kangaroo*. Top: An LLM generates a transparent blending function that evaluates if (i) there exists an enemy close to the player and if (ii) nothing is around the player. Bottom: Two states are evaluated: the body predicates (*i.e.* `nothing_around` and `closeby`) values are multiplied with the rules' associated learnable weight (displayed on their left), to obtain the neural/logic importance weighting.

These rules are transparent, *e.g.* [R1] can be interpreted as "Select `UP` if the oxygen is empty". The body atom `is_empty` is a *state predicate*, whose truth level can be computed from an object-centric state. Each state predicate is associated with a

```
[R1] 0.73: up(X):-is_empty(Oxygen).
[R2] 0.42: up(X):-above(Diver,Agent).
[R3] 0.31: left(X):-left_of(Diver,Agent).
```
Listing 1: Exemplary action rules for *Seaquest*.

(differentiable) function, known as a *valuation function*, to compute its truth value, or confidence. For example, `is_empty` can be mapped to a function, such as $sigmoid((x - \alpha)/\gamma)$, which translates the actual oxygen value $x \in [0, 100]$ (from the object-centric state) into a truth score ranging from $[0, 1]$. The second rule [R2] represents the same action selection, `UP`, but is motivated by the aim of collecting divers. The third rule [R3] selects another action (`LEFT`), if a diver is left of the player. After evaluating the valuation functions for each state predicate, we perform differentiable forward reasoning (Shindo et al., 2023) to deduce action atoms defined by the action rules based on the state atoms. Forward reasoning involves inferring all deducible knowledge (*i.e.* the head atoms for actions) from an observed state. This process allows us to obtain confidence levels (as probabilities) for all actions defined in the symbolic policy.

Contrary to NUDGE policies (Delfosse et al., 2023a), BlendRL generates action rules and their necessary elements (predicates and their valuation functions) using a Large Language Model (LLM), as described in Section 3.3. We also integrate memory-efficient message-passing forward reasoners (Shindo et al., 2024b) to overcome potential memory bottlenecks associated with conventional symbolic policies. NUDGE and other common logic policies consumes memory quadratically with respect to the number of relations and entities, significantly limiting their scalability. In contrast, BlendRL's symbolic policy scales linearly, suitable to more complex environments scalable training parallelization.

**The blending module** is a differentiable function parameterized by $\lambda$. It can be encoded as an explicit logic-based function ($B_\lambda : \mathbb{R}^{F \times n \times m} \to [0, 1]$), as an implicit neural network-based (*i.e.* pixel) state evaluator ($B_\lambda : \mathbb{R}^{F \times W \times H \times C} \to [0, 1]$), or a combination of both. While a logic-based policy provides the advantage of transparency, it relies on inductive biases. If these are not available, a neural blender is necessary.

Figure 3 describes the overall process of the blending module. It computes distributions over neural and logic policies given symbolic states, based on the blending rules generated by LLMs. The blending weighted rule set of an agent trained on *Kangaroo* is depicted in the top right corner. It encodes the fact that the neural module should be selected when a monkey or a deadly thrown coconut is around (to allow for dodging the coconut or adjusting the position to optimally punch the monkey). When nothing is around the agent, it can safely rely on its logic policy (depicted just above it), which allows it to navigate the *Kangaroo* environment. Formally, the blending module provides the weight $\beta \in [0, 1]$ of the neural policy, $\beta = B_\lambda(z)$ or $\beta = B_\lambda(x)$, (and thus implicitly of the neural policy). We empirically show on *Kangaroo* and

```
% Policy ruleset (Kangaroo)
0.73 up(X):-on_ladder(Player,Ladder),same_floor(Player,Ladder).
0.74 right(X):-left_of(Player,Ladder),same_floor(Player,Ladder).
0.72 left(X):-right_of(Player,Ladder),same_floor(Player,Ladder).
...
% Blending ruleset (Kangaroo)
0.78 neural(X):-closeby(Player,Monkey).
0.51 neural(X):-closeby(Player,Coconut).
0.11 logic(X):-nothing_around(Player).
```

Figure 4: **BlendRL provides interpretable policies and blending modules as sets of weighted first-order logic rules.** A subset of the policy's weighted action rules and the blending module's rules from a BlendRL agent trained on *Kangaroo*. All the logic rule sets for each environment are provided in Appendix A.5.

*Seaquest* BlendRL agents with symbolic blending modules outperform ones with neural modules (*cf.* Table 8 in Appendix A.10).

Finally, given a raw state $x$, and its symbolic version, $z$, the final action distribution is computed as:

$$\pi_{(\theta,\phi,\lambda)}(s) = \beta \cdot \pi_\theta^{neural}(x) + (1 - \beta) \cdot \pi_\phi^{logic}(z), \tag{1}$$

where $s = (x, z)$. Note that the blending module can also become a rigid selector, if replaced by $\mathbb{1}_{\beta > 0.5}$.

### 3.3 LLM GENERATED LOGIC POLICIES

BlendRL employs Language Models (LLMs) to generate symbolic programs for precise reasoning, following a chain of thought principle (Wei et al., 2022; Kojima et al., 2022):

**(i)** it uses the task context and detectable objects description and implementation to create state predicates,

**(ii)** it formulates action rules, with conjunctions of the generated predicates as body,

**(iii)** it generates the predicates' *valuations* (*i.e.* their python implemented functions).

For steps **(ii)** and **(iii)**, we used the few-shot prompting approach, providing the LLM with an example logic ruleset obtained from NUDGE policies (Delfosse et al., 2023a). This circumvents the need for an expert to provide the set of logic rules, the used logic predicates, and their implementations, thus allowing users to effectively introduce inductive biases in natural language (*cf.* Appendix A.4 and A.5 for additional details). A subset of the logic module of a BlendRL trained on *Kangaroo* is provided in Figure 4. The associated weights of the LLM-generated rules have been adjusted to maximize performance. GPT4-o[3] was the chosen LLM that we consistently used in our experiments.

### 3.4 OPTIMIZATION

We use the PPO actor-critic to train BlendRL agents, and also employ hybrid value functions. We compose the hybrid critics by computing values using both visual and object-centric states.

**Mixed Value Function.** As the value function approximates the expected return, we did not use logic to encode it. However, BlendRL incorporates a hybrid value function, that uses both the subsymbolic ($x$) and symbolic (or object-centric $z$) state representations. Given state $s = (x, z)$, the value is defined as:

$$V_{(\mu,\omega)}(s) = \beta \cdot V_\mu^{CNN}(x) + (1 - \beta) \cdot V_\omega^{OC}(z). \tag{2}$$

with $V_\mu^{CNN}$ a CNN, that share its convolutional layers with the neural actor (Schulman et al., 2017) and $V_\omega^{OC}$ a small MLP on the object-centric state, and $\beta = B_\lambda(x, z)$, the blending weight.

---

[3] https://openai.com/index/hello-gpt-4o/

| Environments | DQN | PPO | NLRL | NUDGE | SCoBots | INTERP. | INSIGHT | BlendRL |
|---|---|---|---|---|---|---|---|---|
| **Kangaroo** | 98.4 | 26.1 | 111.2 | 112.1 | 149.7 | 64.4 | - | **474.4** |
| **Seaquest** | 61.3 | 14.8 | -3.3 | -3.6 | 52.2 | 20.8 | 58.2 | **94.7** |
| **DonkeyKong** | 3.5 | 28.4 | 0.4 | 1.7 | 5.8 | 25.1 | - | **48.4** |

Table 1: **BlendRL surpasses deep and symbolic agents in our evaluation.** Human normalized scores of BlendRL to different deep (DQN and PPO) and symbolic (NLRL, NUDGE, SCoBots, INTERPRETER and INSIGHT) baselines. Raw results for each agent type, as well as human scores are given in Appendix A.6.

**Regularization.** To ensure the use of both neural and logic policies, we introduce a regularization term for BlendRL agents. To further enhance exploration, we penalize overly peaked blending distributions. This approach ensures that agents utilize both neural and logic policies without deactivating either entirely.

$$R = -\beta \log \beta - (1 - \beta) \log(1 - \beta) \tag{3}$$

This helps agents avoid suboptimal policies typically caused by neural policies due to the sparse rewards.

## 4 EXPERIMENTS

We outline the benefits of BlendRL over purely neural or symbolic approaches, supported by additional investigations into BlendRL's robustness to environmental changes. Furthermore, we examine the interactions between neural and symbolic components and demonstrate that BlendRL can generate faithful explanations. We specifically aim to answer the following research questions:
**(Q1)** Can BlendeRL agents overcome both symbolic and neural agents' shortcomings?
**(Q2)** Can BlendRL can produce both neural and symbolic explanations for its action selection?
**(Q3)** Are BlendRL agents robust to environmental changes?
**(Q4)** How do the neural and symbolic modules interact to maximize BlendRL's agents overall performances?

Let us now provide empirical evidence for BlendRL's ability to learn efficient and understandable policies, even without being provided with all the necessary priors to optimally solve the tasks.

### 4.1 EXPERIMENTAL SETUP

**Environments.** We evaluate BlendRL on Atari Learning Environments (Bellemare et al., 2013), the most popular benchmark for RL (particularly for relational reasoning tasks). For resource efficiency, we use the object centric extraction module of (Delfosse et al., 2024b). Specifically, in *Kangaroo*, the agent needs to reach and climb ladders, leading to the captive joey, while punching incoming monkey antagonists, that try to stop it. In *Seaquest*, the agent has to rescue divers, while shooting sharks and enemy submarines. It also needs to surface before the oxygen runs out. Finally, in *Donkey Kong* the agent has to reach the princess at the top, while avoiding incoming barrels thrown by Donkey Kong. For additional details, *cf.* Appendix A.9. To further test BlendRL abilities to overcome the potential lack of concept necessary to solve task, we omitted some part of the game in our prompt for the LLM that generates the policy rules. Specifically, we kept out the facts that agents can punch the monkeys in *Kangaroo*, can shoot the enemies in *Seaquest* and can jump over the barrels in *DonkeyKong*. To test robustness and isolate different abilities of the agents, we employ HackAtari (Delfosse et al., 2024a), that allow to customize the environments (*e.g.* remove enemies). For these ablation studies, we provide details about the use modification at the relevant point of the manuscript.

**Baselines.** We compare BlendRL to purely neural PPO and DQN agents. Both agent types incorporate the classic CNN used for Atari environments. Additionally, we evaluate NUDGE, which uses a pretrained neural PPO agent for searching viable policy rules (Delfosse et al., 2023a), and NLRL (Jiang & Luo, 2019), which

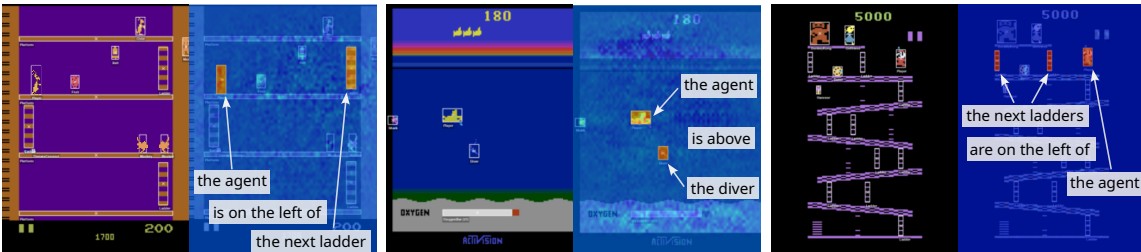

Figure 5: **Qualitative examples of explanations produced by BlendRL,** in the form of importance maps, computing with gradient-based attributions for both neural and logic policies. The logic-based module also allows BlendRL to generate textual interpretations, enhancing its transparency and interoperability. Inspired by (Luo et al., 2024), we generate textual explanations using LLMs, that we provide in App. A.15.

generates rules based on templates Evans & Grefenstette (2018). We also provide the reported scores of the following (neuro-)symbolic agents: SCoBots (Delfosse et al., 2024d), INTERPRETER Kohler et al. (2024) and INSIGHT (Luo et al., 2024) agents (or extend their evaluation if not available, *e.g.* in *DonkeyKong*). We train each agent types until all of them converge to a stable episodic return (*i.e.* for 15K episodes for *Kangaroo* and *DonkeyKong* and 25K for *Seaquest*). For additional details, *cf.* Appendix A.8.

## 4.2 RESULTS AND ANALYSIS

**Comparison to neural and neuro-symbolic agents (Q1).** Table 1 shows the final human normalized scores of each evaluated agent across our selected Atari environments. BlendRL surpasses all the logic-based baselines in all tested scenarios. In the *Kangaroo* environment, which requires fewer intuitive actions due to the relatively small number of enemies, NUDGE shows fair performance, even if its are less able to punch the monkeys and avoid their thrown coconuts. However, in the other environments, populated with more incoming threats, where neural policy becomes critical for accurate controls, NUDGE lags behind. Additionally, the purely neural PPO agents often fall into suboptimal policies. For instance, in *Seaquest*, surfacing without collecting any divers lead to negative reward. Neural PPO agents thus concentrate on shooting sharks to collect reward, but never refill their oxygen. In contrast, BlendRL effectively selects its logic module to collect divers and surface when needed and its neural module to efficiently align itself with the enemies and shoot them. Overall, BlendRL significantly outperforms both baselines across different environments, underscoring the efficacy of neuro-symbolic blending policies in efficiently harnessing both neural and symbolic approaches to policy learning.

**BlendRL agents are interpretable and explainable (Q2).** BlendRL's symbolic policies can easily be interpreted as they consist of a set of transparent symbolic weighted rules, as exemplified in Figure 4 for *Kangaroo*. The interpretable blending module prioritizes neural agents in situations where finetuned control is needed (*i.e.* for dodging an incoming deadly coconut or accurately placing oneself next to the monkey and punching it). Conversely, the logic module is in use when no immediate danger is present, *e.g.* as a path finding proxy. The logic rules for the other environments are provided in Appendix A.5.

BlendRL also produces gradient-based explanations, as each component is differentiable. Logic-based explanations are computed using action gradients (*i.e.* $\partial \pi_\phi^{logic}(\mathbf{z})/\partial \mathbf{z}$) on the state atoms, that underline the most important object properties for the decision. Further, the integrated gradients method (Sundararajan et al., 2017) on the neural module provides importance maps. We can visualize these two explanations, merging them with the blending weight $\beta$, as shown in Figure 5. The most important logic rules for the decision are also depicted. BlendRL provides interpretable rules for it reasoning part and importance maps.

**BlendRL is robust to environmental changes (Q3).** Deep agents usually learn "by heart" spurious correlations during training, and thus fail to generalize to unseen environments, even on simple Atari games (Farebrother et al., 2018; Delfosse et al., 2024a). We used HackAtari variations of the environment, to disable

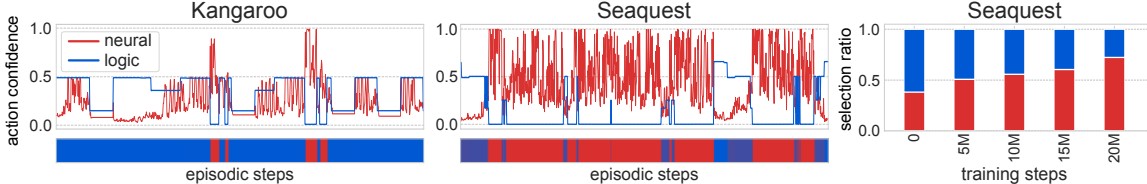

Figure 6: **BlendRL uses both its neural and logic modules.** Through an episode, the highest action probability is depicted (left, top), with the selected module at this step (bottom). BlendRL makes use of both module types through an episode. It progressively modifies their selection ratio along training on Seaquest (right), as its progress allows it to uncover parts of the game with more enemies.

the enemies in *Kangaroo* and *Seaquest* and to disable the barrels in *DonkeyKong*. We additionally used a modified *Kangaroo* environment with relocated ladders.

As expected, our BlendRL agent can still complete the tasks (and thus collect reward) on these safer versions of the environments (*cf.* Table 2). BlendRL agents indeed rely on the knowledge incorporated in their logic-based policies and blending modules, and only rely on their neural modules for control accurate skills (*e.g.* aiming and shooting/punching), as we further show hereafter.

| Return | Kangaroo | Seaquest | DonkeyKong |
|---|---|---|---|
| Neural PPO | $0.0_{\pm 0.0}$ | $0.0_{\pm 0.0}$ | $0.0_{\pm 0.0}$ |
| BlendRL | $\mathbf{187_{\pm 120}}$ | $\mathbf{113_{\pm 106}}$ | $\mathbf{263_{\pm 124}}$ |

Table 2: **Contrary to neural PPO, BlendRL is robust to environmental changes.** Average returns on the modified environments (over 10 seeds).

**Neural and symbolic policy interactions in BlendRL (Q4).** We here investigate how the symbolic and neural policies interoperate. One concern of such an hybrid systems is its potential reliance on only one module. The system could indeed learn to *e.g.* only rely on its neural component (*i.e.* $\beta \approx 1.0$). Figure 6 (left and center) illustrates the outputs of BlendRL's neural and logic policies for $1k$ steps (*i.e.* 1-3 episodes). Specifically, the maximum value of the action distributions ($\max_a \pi_\phi^{logic}(a|s_t)$ and $\max_a \pi_\theta^{neural}(a|s_t)$), at each time step $t$ is depicted (at the top), underlined by the importance weighting of each module (*blue* if logic is mainly selected ($\beta \approx 0$), *red* if neural is ($\beta \approx 1$)), indicating how much each module is used at each step on *Kangaroo* and *Seaquest*. Further, BlendRL's agents can adapt the amount to which they select each component through training, as depicted on the right of this figure. As *Seaquest* is a progressive environment, in which the agent first faces few enemies (thus mainly relying on its logic module to collect the divers), before progressively accessing states in which many more enemies are spawning (*cf.* Figure 12), BlendRL agents first mainly relies on its logic module (blue), while progressively shifting this preference to its neural one (red) for accurately shooting enemies. These results demonstrate the efficacy of BlendRL's policy reasoning and learning on neuro-symbolic hybrid representations, thereby enhancing overall performance. We provide a further analysis comparing neural and logic blending modules in Appendix A.10, highlighting that the logic-based blending module can utilize both policies effectively, resulting in better performance.

Overall, our experimental evaluation has shown BlendRL's agents ability to learn on several Atari environments that require both reasoning and reacting capabilities. We showed that they outperform the commonly used neural PPO baseline, as well as the state-of-the-art logic agents, NUDGE. We further demonstrated their ability to generalize to unseen scenarios, on slightly modified environments (compared to the training ones), and that they can efficiently alternate between the two module types to obtain policies that can both produce interpretations, quantifying the impact of each symbolic properties and of each pixel region.

## 5 RELATED AND FUTURE WORK

Relational Reinforcement Learning (Relational RL) (Dzeroski et al., 2001; Kersting et al., 2004; Kersting & Driessens, 2008; Lang et al., 2012; Hazra & Raedt, 2023) has been developed to address RL tasks in

relational domains by incorporating logical representations and probabilistic reasoning. BlendRL extends this approach by integrating differentiable logic programming with deep neural policies. The Neural Logic Reinforcement Learning (NLRL) framework (Jiang & Luo, 2019) was the first to incorporate Differentiable Inductive Logic Programming ($\partial$ILP) (Evans & Grefenstette, 2018) into the RL domain. $\partial$ILP learns generalized logic rules from examples using gradient-based optimization. NUDGE (Delfosse et al., 2023a) extends this method by introducing neurally-guided symbolic abstraction, leveraging extensive work on $\partial$ILP (Shindo et al., 2021b;a) to learn complex programs. INSIGHT (Luo et al., 2024) is another neurosymbolic framework that learns object-centric states and neural policies, distilled in EQL-based policies, then provided to an LLM, that produces textual explanations about these policies. These approaches aim to represent policies using symbolic representations. In contrast, BlendRL integrates both neural and symbolic policies and leverages both during inference.

The integration of planners with RL has been explored to achieve deliberate thinking in policy learning. For example, RePReL (Kokel et al., 2021) decomposes multi-agent planning tasks using a planner and then employs reinforcement learning for each agent to solve subtasks. In these frameworks, planners for longterm (slow) reasoning are typically separate components. In contrast, BlendRL computes both symbolic and neural policies at the same level, allowing them to be learned jointly, thereby enhancing overall performance. Additionally, planners are often used in model-based RL to generate hypothetical experiences that improve value estimates (Sutton, 1991; Kaiser et al., 2019). In contrast, BlendRL incorporates the symbolic reasoning module directly into its policy, enabling joint learning with neural modules. Another notable approach is to use Linear Temporal Logic (LTL) to instruct RL agents (León et al., 2022; Kuo et al., 2020; Vaezipoor et al., 2021; Qiu et al., 2023), where LTL formulas represent instructions to guide agents and the task is to train (deep) agents to follow them. In contrast, in BlendRL, logic expresses the policy itself and directly influences decision-making. To this end, various interpretations of the *fast and slow* systems have been developed for RL (Botvinick et al., 2019; Duan et al., 2017; Tan & Motani, 2023; Anthony et al., 2017). While deep agents focused on object-centric and relational concepts have been explored (Shanahan et al., 2020; Feng & Magliacane, 2023), BlendRL uniquely integrates these concepts with hybrid neuro-symbolic policies, making it applicable to environments that require both abstract reasoning and reactive actions.

Additionally, BlendRL is related to prior work using LLMs for program generation. For example, LLMs have been applied to generate probabilistic programs (Wong et al., 2023), Answer Set Programs (Ishay et al., 2023; Yang et al., 2023), differentiable logic programs (Shindo et al., 2024a), and programs for visual reasoning (Surís et al., 2023; Stanić et al., 2024). Our symbolic policy representation is inspired by *situation calculus* (Reiter, 2001), which is an established framework to describe states and actions in logic.

## 6 CONCLUSION

In this study, we introduced BlendRL, a pioneering framework that integrates symbolic and neural policies for reinforcement learning. BlendRL employs neural networks for reactive actions and differentiable logic reasoners for high-level reasoning, seamlessly merging them through a blending module that manages distributions over both policy types. We also developed a learning algorithm for BlendRL agents that hybridizes the state-value function on both pixel-based and object-centric states and includes a regularization approach to enhance the efficacy of both logic and neural policies.

Our empirical evaluations demonstrate that BlendRL agents significantly outperform purely neural agents and state-of-the-art neuro-symbolic baselines in popular Atari environments. Furthermore, these agents exhibit robustness to environmental changes and are capable of generating clear, interpretable explanations across various types of reasoning, effectively addressing the limitations of purely neural policies. Our comprehensive analysis of the interactions between symbolic and neural policy representations highlights their synergistic potential to enhance overall performance.

**Acknowledgements.** The authors thank Sriraam Natarajan for his valuable feedback on the manuscript. This research work was partly funded by the German Federal Ministry of Education and Research, the Hessian Ministry of Higher Education, Research, Science and the Arts (HMWK) within their joint support of the National Research Center for Applied Cybersecurity ATHENE, via the "SenPai: XReLeaS" project. It also benefited from the HMWK Cluster Project 3AI. We gratefully acknowledge support by the EU ICT-48 Network of AI Research Excellence Center "TAILOR" (EU Horizon 2020, GA No 952215), and the Collaboration Lab "AI in Construction" (AICO). The Eindhoven University of Technology authors received support from their Department of Mathematics and Computer Science and the Eindhoven Artificial Intelligence Systems Institute.

**Ethics Statement.** This research aims to advance neuro-symbolic reinforcement learning through the BlendRL framework, which combines neural and symbolic policies for enhancing performance and transparency. Our work includes no harmful examples or applications and is intended solely for positive contributions to AI and machine learning. We conducted our experiments using well-established benchmark environments, specifically popular Atari games, which are ethically recognized testbeds in the research community. Our methods and results aim to advance the field responsibly. Our code and resources are openly available to ensure transparency, reproducibility, and to foster further research in this area.

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

# A APPENDIX

## A.1 DETAILS OF DIFFERENTIABLE FORWARD REASONING

In this section, we provide the details of differentiable forward reasoning.

### A.1.1 WHAT IS FORWARD REASONING?

*Forward Reasoning* is a data-driven approach of reasoning in FOL (Russell & Norvig, 2010). Forward reasoning is performed by applying a function called the $T_{\mathcal{C}}$ *operator*, deducing new ground atoms using given clauses and ground atoms.

For a set of clauses $\mathcal{C}$, $T_{\mathcal{C}}$ operator (Lloyd, 1984) is a function that applies clauses in $\mathcal{C}$ using given ground atoms $\mathcal{G}$, *i.e.*

$$T_{\mathcal{C}}(\mathcal{G}) = \mathcal{G} \cup \left\{ A \mid \begin{array}{l} A \text{:-} B_1, \ldots, B_n \in \mathcal{C}^* \\ (\{B_1 \ldots, B_n\} \subseteq \mathcal{G}) \end{array} \right\}, \tag{4}$$

where $\mathcal{C}^*$ is a set of all ground clauses that can be produced from $\mathcal{C}$.

**Example.** Let $\mathcal{C}$ be the following rules:

```
% Policy ruleset (Kangaroo)
up(X):-on_ladder(Player,Ladder),same_floor(Player,Ladder).
right(X):-left_of(Player,Ladder),same_floor(Player,Ladder).
left(X):-right_of(Player,Ladder),same_floor(Player,Ladder).
```

First, the rules are grounded by considering all possible substitutions to the variables. For example, we have the following grounded rules:

```
% Policy ruleset (Kangaroo)
up(img):-on_ladder(player,ladder1),same_floor(player,ladder1).
right(img):-left_of(player,ladder1),same_floor(player,ladder1).
left(img):-right_of(player,ladder1),same_floor(player,ladder1).
```

For simplicity, we consider 3 constants: `img`, `player`, `ladder1`, and generate only grounding that satisfies type constraints, *e.g.* we do not generate $\text{on\_ladder}(\text{ladder1}, \text{ladder1})$ as it violates the type specification of the predicate, *i.e.*, `on_ladder` should take constants that represent the player and a ladder as arguments. Now, following Eq. 4, the head atoms that represent actions are deduced by applying the $T_{\mathcal{C}}$ operator:

In BlendRL, the initial set of ground atoms $\mathcal{G}^{(1)}$ is a set of state atoms that encodes an observed state. Let $\mathcal{G}^{(1)}$ be a set of ground atoms that contain the following ground atoms:

```
on_ladder(player,ladder1),same_floor(player,ladder1),
left_of(player,ladder1),right_of(player,ladder1).
```

Then $\mathcal{G}^{(2)} = T_{\mathcal{C}}(\mathcal{G}^{(1)})$ is computed as follows:

```
up(img),left(img),right(img),
on_ladder(player,ladder1),same_floor(player,ladder1),
left_of(player,ladder1),right_of(player,ladder1).
```

where the first 3 ground atoms are newly deduced by 1-step forward reasoning. In this example, all rules are considered true (1.0 of weight for each), and 3 different actions of `up`, `left`, `right` are deduced from an observed state. By using $\mathcal{G}^{(2)}$, the agent can perform action decisions based on logical deductive reasoning.

Now let us move on how to perform forward reasoning in a differentiable manner.

### A.1.2 WHAT IS DIFFERENTIABLE FORWARD REASONING?

*Differentiable forward reasoning* has been proposed by (Evans & Grefenstette, 2018). It encodes the forward-chaining reasoning process in FOL using differentiable tensor operations. To mitigate its memory-consuming bottleneck (*cf.* subsequent subsection A.1.3 for details), using graph representations instead of tensors has been proposed (Shindo et al., 2024b). We adapt this approach to the RL domain.

The key idea is to represent a set of rules as a bipartite graph and perform message passing to conduct forward reasoning. Let us introduce *forward reasoning graph*, a core concept of this approach.

**Definition A.1** *A Forward Reasoning Graph is a bipartite directed graph* $(\mathcal{V}_{\mathcal{G}}, \mathcal{V}_{\wedge}, \mathcal{E}_{\mathcal{G}\to\wedge}, \mathcal{E}_{\wedge\to\mathcal{G}})$, *where* $\mathcal{V}_{\mathcal{G}}$ *is a set of nodes representing ground atoms (atom nodes),* $\mathcal{V}_{\wedge}$ *is set of nodes representing conjunctions (conjunction nodes),* $\mathcal{E}_{\mathcal{G}\to\wedge}$ *is set of edges from atom to conjunction nodes and* $\mathcal{E}_{\wedge\to\mathcal{G}}$ *is a set of edges from conjunction to atom nodes.*

**Example (reasoning graph).** Fig 7 depicts the forward reasoning graph for the Kangaroo policy rules in Fig. 4. For simplicity, we visualize for the first 2 rules that define actions of `up` and `right`.

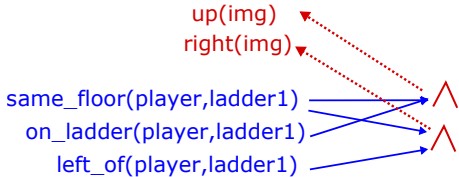

Figure 7: Forward reasoning graph for policy rules for Kangaroo. A reasoning graph consists of *atom nodes* (left) and *conjunction nodes* (right). Only relevant nodes are shown (Best viewed in color).

A reasoning graph is represented as a bipartite graph. The left nodes represent ground atoms, and the right nodes represent conjunctions representing bodies of ground rules, *i.e.* , a conjunction node corresponds to a ground clause. It is obtained by grounding rules *i.e.* , removing variables by, *e.g.* , `Player ← player`, `Ladder ← ladder1`. By performing bi-directional message passing on the reasoning graph using soft-logic operations, BlendRL computes logical consequences in a differentiable manner.

Let us move on to how to perform message passing to perform forward reasoning. Essentially, forward reasoning consists of *two* steps: (1) computing conjunctions of body atoms for each rule and (2) computing disjunctions for head atoms deduced by different rules. These two steps can be efficiently computed on bi-directional message-passing on the forward reasoning graph. We now describe each step in detail.

**(Direction →) From Atom to Conjunction.** First, messages are passed to the conjunction nodes from atom nodes. For conjunction node $v_i \in \mathcal{V}_{\wedge}$, the node features are updated:

$$v_i^{(t+1)} = \bigvee \left( v_i^{(t)}, \bigwedge_{j \in \mathcal{N}(i)} v_j^{(t)} \right), \tag{5}$$

where $\bigwedge$ is a soft implementation of *conjunction*, and $\bigvee$ is a soft implementation of *disjunction*. Intuitively, probabilistic truth values for bodies of all ground rules are computed softly by Eq. 5.

**(Direction ←) From Conjunction to Atom.** Following the first message passing, the atom nodes are then updated using the messages from conjunction nodes. For atom node $v_i \in \mathcal{V}_{\mathcal{G}}$, the node features are updated:

$$v_i^{(t+1)} = \bigvee \left( v_i^{(t)}, \bigvee_{j \in \mathcal{N}(i)} w_{ji} \cdot v_j^{(t)} \right), \tag{6}$$

where $w_{ji}$ is a weight of edge $e_{j \to i}$. We assume that each rule $C_k \in \mathcal{C}$ has its weight $\theta_k$, and $w_{ji} = \theta_k$ if edge $e_{j \to i}$ on the reasoning graph is produced by rule $C_k$. Intuitively, in Eq. 6, new atoms are deduced by gathering values from different ground rules and from the previous step.

We used product for conjunction, and *log-sum-exp* function for disjunction:

$$softor^\gamma(x_1, \ldots, x_n) = \gamma \log \sum_{1 \le i \le n} \exp(x_i/\gamma), \tag{7}$$

where $\gamma > 0$ is a smooth parameter. Eq. 7 approximates the maximum value given input $x_1, \ldots, x_n$.

**Example (message passing).** Fig. 8 illustrates the message-passing process on the Kangaroo reasoning graph. We consider an observed object-centric state such that state atoms have the following distribution:

```
0.9: same_floor(player,ladder1)
0.3: on_ladder(player,ladder1)
0.8: right_of(player,ladder1)
```

(Direction $\to$) These values (weights of $[0.9, 0.3, 0.8]$) are assigned to corresponding atom nodes, and these values are passed to the conjunction nodes following Eq. 5 (*cf.* the left-most figure in Fig. 8).

(Direction $\leftarrow$) A smooth operation amalgamates the incoming values to conjunction nodes. We simply employ the arithmetic multiplication here. *I.e* for each conjunction node, we compute the multiplication of all incoming messages. The resulting values correspond to evaluation scores of bodies of ground rules. Then, the values on conjunction nodes are passed to the left for action atoms (*cf.* Fig. 8).

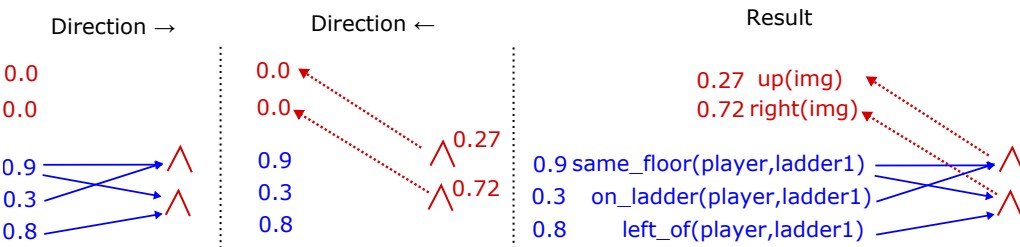

Figure 8: Message passing on the reasoning graph. (Best viewed in color).

**Prediction.** The probabilistic logical entailment is computed by the bi-directional message-passing. Let $\mathbf{x}_{atoms}^{(0)} \in [0,1]^{|\mathcal{G}|}$ be input node features, which map a fact to a scalar value, RG be the reasoning graph, $\mathbf{w}$ be the rule weights, $\mathcal{B}$ be background knowledge, and $T \in \mathbb{N}$ be the infer step. For fact $G_i \in \mathcal{G}$, BlendRL computes the probability as:

$$p(G_i \mid \mathbf{x}_{atoms}^{(0)}, \mathsf{RG}, \mathbf{w}, \mathcal{B}, T) = \mathbf{x}_{atoms}^{(T)}[i], \tag{8}$$

where $\mathbf{x}_{atoms}^{(T)} \in [0,1]^{|\mathcal{G}|}$ is the node features of atom nodes after $T$-steps of the bi-directional message-passing.

**Example (prediction).** In the right-most figure in Fig. 8, BlendRL extracts values for action atoms, *i.e.* $[0.27, 0.72]$ for actions `up` and `right`. By taking softmax over these values, BlendRL's symbolic policy computes valid action distributions.

### A.1.3 WHY GRAPH-BASED REASONING?

The original tensor-based reasoner (Evans & Grefenstette, 2018), used in NUDGE (Delfosse et al., 2023a), is memory-intensive. The memory consumption increases quadratically with the number of ground representations of rules and atoms when using tensors, while it increases linearly with graph representations.

Let $C$ be the number of rules in the policy and $G$ be the number of atoms representing actions and states. We can derive the number of ground representations of these $C^*$ and $G^*$, respectively. Note that $C^*$ and $G^*$ can be very large as they reflect all possible groundings (substitutions to remove variables in the rules and atoms). The tensor-based reasoner consumes quadratically $\mathcal{O}(C^* \times G^*)$. However, the graph-based reasoner consumes only linearly $\mathcal{O}(C^* + G^*)$.

This greatly affects the scalability of the method. In practice, we consider parallelized (vectorized) environments to train agents to get diverse experiences. Let $N_{env}$ be the number of parallelized environments to train agents. Then the tensor-based consumes memory $\mathcal{O}(N_{env} \times C^* \times G^*)$ but the graph-based one consumes only $\mathcal{O}(N_{env} \times (C^* + G^*))$. We empirically observed that memory efficiency is the key to BlendRL agents' successful training. In Seaquest, the tensor-based one scaled up to most 100 environments on NVIDIA A100-SXM4-40GB GPU. However, the graph-based one scaled to more than 500 parallelized environments, significantly improving performance.

It has been proven that the graph-based reasoner can handle complex programs with function symbols (Shindo et al., 2024b). Consequently, BlendRL is capable of handling programs that contain function symbols due to its memory-efficient graph-based reasoner. Incorporating programs with structured terms using function symbols, such as lists and trees, would be a significant enhancement e.g. meta interpreters[4].

### A.2 RULE GENERATION DETAILS

In this section, we present the complete set of prompts utilized for our rule generation process. We consistently employed GPT-4o [5] for generating these rules.

Initially, we provided a general format instruction as a prompt, detailing the structure of rules that define actions. The prompt used was as follows:

```
Given a instruction and available predicates, generate rules that define actions in the rule format.
The rule format is
    action(X):-cond_1,...,cond_n.
Each cond_i can be either of:
   (1) pred(Object)
   (2) pred(Player)
   (3) pred(Player,Object)
where "action" and "pred" are provide predicates. Use predicates provided as available_predicates.
    Show only the rule without any decoration.
```

Subsequently, we provide examples demonstrating rule generation. All the rules included here are derived from a trained NUDGE agent on the GetOut environment (Delfosse et al., 2023a), which is a non-Atari environment. We have added textual interpretations for each rule, pairing them with the corresponding output.

```
Examples:

Jump if an enemy is getting close by the player.
jump(X):-closeby(Player,Enemy).

Go left to get the key if the player doesn't have the key and the player is on the right of the key.
left_to_get_key(X):-not_have_key(Player),on_right(Player,Key).
```

---

[4] https://www.metalevel.at/acomip
[5] https://openai.com/index/hello-gpt-4o/

```
Go left to go to the door if the player has the key and the player is on the right of the door.
left_to_door(X):-have_key(Player),on_right(Player,Door).

Go right to get the key if the player doesn't have the key and the player is on the left of the key.
right_to_get_key(X):-not_have_key(Player),on_left(Player,Key).

Go right to go to the door if the player has the key and the player is on the left of the door.
right_to_door(X):-have_key(Player),on_right(Player,Door).
```

Finally, we offer environment-specific instructions, detailing the complete prompt of actions for each environment:

```
% Kangaroo and DonkeyKong
Go up if the player is on the ladder and the player and the ladder are on the same level.
Go right to the ladder if the player is on the left of the ladder and the player and the ladder are on
    the same level.
Go left to the ladder if the player is on the right of the ladder and the player and the ladder are on
    the same level.

% Seaquest
Go up to the diver if the player is deeper than the diver, the diver is visible, and the collected
    divers are not full.
Go up to rescue if full divers are collected.
Go left if the player is on the right of the diver, the diver is visible, and the collected divers are
    not full.
Go right if the player is on the left of the diver, the diver is visible, and the collected divers are
    not full.
Go up to the diver if the player is deeper than the diver, the diver is visible, and the collected
    divers are not full.
Go down to the diver if the player is higher than the diver, the diver is visible, and the collected
    divers are not full.
```

For the blending rules, we used the following prompts:

```
% Kangaroo
If a monkey or coconut is close by, use the neural agent.
If nothing is around the player, use the logic agent.
% Seaquest
If a shark or missile is close by, use the neural agent.
If nothing is around the player, use the logic agent.
% DonkeyKong
If a barrel is close by, use the neural agent.
If nothing is around the player, use the logic agent.
```

## A.3 PREDICATE GENERATION DETAILS

In this section, we provide detailed explanations on the predicate (relation) generations for the relational state representations in BlendRL. We consistently employed GPT-4o [6] for generating these rules.

Initially, we provided general task instructions of the predicate generation task as a prompt:

```
Given a task concept and an object description, generate a set of predicates (relations) to represent
    the state of the task. Use only simple concepts as predicates.
```

Subsequently, we provide examples demonstrating predicate generation. All the environments included here (GetOut, 3Fish, and Heist) are environments used for NUDGE Delfosse et al. (2023a), which are non-Atari environments. We have added textual task descriptions for each environment, pairing them with the corresponding output.

---

[6] https://openai.com/index/hello-gpt-4o/

```
Example:
In the GetOut environment, the task is to go to the door after getting a key. You will find enemies
    and need to dodge them when they get close by.
That means if the key is on the left of the player, they should go right to get the key. On the
    contrary, if the key is on the right of the player, they should go left to get the key.
The same applies to the door.

Predicates:
left_of, right_of, close_by.

Example:
In the 3Fish environment, the task is to survive and grow by eating other fish. You will find enemies
    and need to dodge them when they get close by. That means if a fish that is smaller than the
    player is on the left of the player, they should go right to eat the fish. The same applies to
    other orientations: go left, go above, and go down.

Predicates:
bigger_than, left_of, right_of, above, below

Example:
In the Heist environment, the task is to open boxes by collecting keys. Each box has different colors
    and can be opened only by a key that has the same color. That means if the key is on the left of
    the player, they should go right to get the key. On the contrary, if the key is on the right of
    the player, they should go left to get the key. Also, if the player has a key, they should go to
    the box that has the same color.

Predicates:
left_of, right_of, above, below, same_color
```

Finally, we offer environment-specific instructions, detailing the complete prompt of actions for each environment:

```
% Kangaroo
In the Kangaroo environment, the task is to go to reach the joey. You will find monkeys and need to
    dodge or fire them when they get close by. To reach the joey, the player needs to climb ladders
    to go to the upper floors. That means if the player is on the left of the ladder, they should go
    right to get to the ladder. On the contrary, if the player is on the right of the ladder, they
    should go left to get to the ladder. To climb the ladder, the player should be located on a
    ladder on the same floor.

Predicates:

% Seaquest
In the Seaquest environment, the task is to rescue divers while managing oxygen levels. You will find
    enemy submarines and need to shoot or dodge them when they get close by. That means if a player
    is on the left of the diver, they should go right to rescue the diver. On the contrary, if a
    player is on the right of the diver, they should go left to rescue the diver. This applies to
    other directions: above and below. If divers are fully collected (6 divers), then go up to the
    surface to rescue them. Additionally, the player must monitor oxygen levels and ascend to the
    surface to replenish when low.

Predicates:

% DonkeyKong
In the Donkey Kong environment, the task is to reach the top, where the player can rescue the
    character while avoiding obstacles. You will find barrels rolling towards the player and need to
    jump over them when they get close by. That means if the player is on the left of the ladder,
    they should go right to reach the ladder. On the contrary, if the player is on the right of the
    ladder, they should go left to reach the ladder. If the player reaches a ladder, they should
    climb it to advance to higher levels.
Predicates:
```

### A.4 BODY PREDICATES AND THEIR VALUATIONS

We here provide examples of valuation functions for evaluating state predicates (*e.g.* `closeby`, `left_of`, etc.) generated by LLMs (in Python).

```python
# A subset of valuation functions for Kangaroo and DonkeyKong (generated by LLMs)
def left_of(player: th.Tensor, obj: th.Tensor) -> th.Tensor:
    x = player[..., 1]
    obj_x = obj[..., 1]
    obj_prob = obj[:, 0] # objectness
    return sigmoid(alpha < obj_x - x) * obj_prob * same_level_ladder(player, obj)

def _close_by(player: th.Tensor, obj: th.Tensor) -> th.Tensor:
    player_x = player[:, 1]
    player_y = player[:, 2]
    obj_x = obj[:, 1]
    obj_y = obj[:, 2]
    obj_prob = obj[:, 0] # objectness
    x_dist = (player_x - obj_x).pow(2)
    y_dist = (player_y - obj_y).pow(2)
    dist = (x_dist + y_dist).sqrt()
    return sigmoid(dist) * obj_prob

def on_ladder(player: th.Tensor, obj: th.Tensor) -> th.Tensor:
    player_x = player[..., 1]
    obj_x = obj[..., 1]
    return sigmoid(abs(player_x - obj_x) < gamma)
...

# A subset of valuation functions for Seaquest (generated by LLMs)
def full_divers(objs: th.Tensor) -> th.Tensor:
    divers_vs = objs[:, -6:]
    num_collected_divers = th.sum(divers_vs[:,:,0], dim=1)
    diff = 6 - num_collected_divers
    return sigmoid(1 / diff)

def not_full_divers(objs: th.Tensor) -> th.Tensor:
    return 1 - full_divers(objs)

def above(player: th.Tensor, obj: th.Tensor) -> th.Tensor:
    player_y = player[..., 2]
    obj_y = obj[..., 2]
    obj_prob = obj[:, 0]
    return sigmoid( (player_y - obj_y) / gamma) * obj_prob
...
```

### A.5 BLENDRL RULES

We here provide the blending and action rules obtained by BlendRL on *Seaquest* and *DonkeyKong*.

```prolog
% Blending rulset (Seaquest)
0.98 neural_agent(X):-close_by_enemy(P,E).
0.74 neural_agent(X):-close_by_missile(P,M).
0.02 logic_agent(X):-visible_diver(D).
0.02 logic_agent(X):-oxygen_low(B).
1.0 logic_agent(X):-full_divers(X).
% Policy rulset (Seaquest)
0.21 up_air(X):-oxygen_low(B).
0.22 up_rescue(X):-full_divers(X).
0.21 left_to_diver(X):-right_of_diver(P,D),visible_diver(D).
```

```
0.24 right_to_diver(X):-left_of_diver(P,D),visible_diver(D).
0.23 up_to_diver(X):-deeper_than_diver(P,D),visible_diver(D).
0.22 down_to_diver(X):-higher_than_diver(P,D),visible_diver(D).

% Blending ruleset (Kangaroo, DonkeyKong)
0.92 neural_agent(X):-close_by_barrel(P,B).
0.28 logic_agent(X):-nothing_around(X).
% Policy ruleset (Kangaroo, DonkeyKong)
0.88 up_ladder(X):-on_ladder(P,L),same_floor(P,L).
0.47 right_ladder(X):-left_of(P,L),same_floor(P,L).
0.18 left_ladder(X):-right_of(P,L),same_floor(P,L).
```

## A.6 DETAILED RESULTS

We hereafter provide the final results of our conducted experiments (*i.e.* DQN, PPO, NLRL, NUDGE and BlendRL), as well as of existing symbolic baselines, namely NLRL (Jiang & Luo, 2019), NUDGE (Delfosse et al., 2023a), SCoBots (Delfosse et al., 2024d), INTERPRETER (Kohler et al., 2024) and INSIGHT (Luo et al., 2024) as well as our trained BlendRL agents.

| Environments | DQN | PPO | NLRL* | Human | Random |
|---|---|---|---|---|---|
| **Kangaroo** | 2696 | 790.0±280.8 | 3034±11 | 2739 | 100 |
| **Seaquest** | 2794 | 837.3±46.7 | 75±0 | 4425 | 215.50 |
| **DonkeyKong** | 253.3±45.1 | 2080±1032 | 29±0 | 7320 | |
| | **NUDGE** | **SCoBots** | **INTERPRETER** | **INSIGHT** | **BlendRL** |
| **Kangaroo** | 3058±25 | 4050.0±217.8 | 1800.0±0.0 | - | 12619±132 |
| **Seaquest** | 64±0 | 2411.3±377.0 | 1092.9±154.6 | 2665.7±728.2 | 4204±10 |
| **DonkeyKong** | 122±25 | 426.7±64.3 | 1837.7±459.4 | - | 3541±43 |

Table 3: Raw scores for deep (DQN and PPO) as well as symbolic (NLRL, NUDGE, SCoBots, INTERPRETER, INSIGHT and BlendRL agents).

We also provide the learning curves for the trained PPO, NUDGE, and BlendRL agents in Fig. 9.

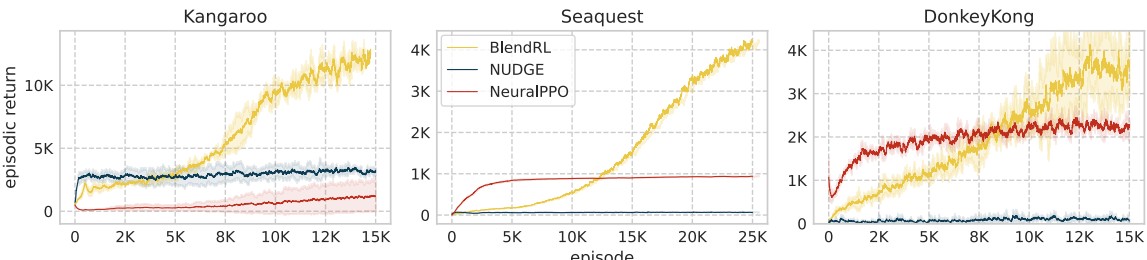

Figure 9: **BlendRL surpasses both purely neural and logic-based policies in environments requiring reasoning and reaction capabilities,** shown by its superior episodic returns after training (averaged over 3 seeded reruns). BlendRL outperforms the logic-based state-of-the-art agent (*i.e.* NUDGE) across all tested environments. The purely neural PPO agents frequently get trapped in suboptimal policies, BlendRL leverages its integrated reasoning and reacting modules to tackle various aspects of the tasks.

### A.7 EXPERIMENTAL DETAILS

We then provide more details about our implementations.

**Hardwares.** All experiments were performed on one NVIDIA A100-SXM4-40GB GPU with Xeon(R):8174 CPU@3.10GHz and 100 GB of RAM.

**Value function update.** We update the value function and the policy as follows: In the following, we consider a (potentially pretrained) actor-critic neural agent, with $v_\phi$ its differentiable state-value function parameterized by $\phi$ (critic). Given a set of action rules $\mathcal{C}$, let $\pi_{(\mathcal{C},\mathbf{W})}$ be a differentiable logic policy. BlendRL learns the weights of the action rules in the following steps. For each non-terminal state $s_t$ of each episode, we store the actions sampled from the policy ($a_t \sim \pi_{(\mathcal{C},\mathbf{W})}(s_t)$) and the next states $s_{t+1}$. We update the value function and the policy as follows:

$$\delta = r + \gamma v_\phi(s_{t+1}) - v_\phi(s_t) \tag{9}$$

$$\phi = \phi + \delta \nabla_\phi v_\phi(s_t) \tag{10}$$

$$\mathbf{W} = \mathbf{W} + \delta \nabla_\mathbf{W} \ln \pi_{(\mathcal{C},\mathbf{W})}(s_t). \tag{11}$$

The logic policy $\pi_{(\mathcal{C},\mathbf{W})}$ thus learns to maximize the expected return.

### A.8 TRAINING DETAILS

We hereby provide further details about the training. Details regarding environments will be provided in the next section A.9. We used the Adam optimizer (Kingma & Ba, 2015) for all baselines.

**BlendRL.** We adopted an implementation of the PPO algorithm from the CleanRL project (Huang et al., 2022). Hyperparameters are shown in Table 4. The object-centric critic is described in Table 5. We provide a pseudocode of BlendRL policy reasoning in Algorithm 1.

| Parameter | Value | Explanation |
|---|---|---|
| $\gamma$ | 0.99 | Discount factor for future rewards |
| learning_rate | 0.00025 | Learning rate for neural modules |
| logic_learning_rate | 0.00025 | Learning rate for logic modules |
| blender_learning_rate | 0.00025 | Learning rate for blending module |
| blend_ent_coef | 0.01 | Entropy coefficient for blending regularization (Eq. 3) |
| clip_coef | 0.1 | Coefficient for clipping gradients |
| ent_coef | 0.01 | Entropy coefficient for policy optimization |
| max_grad_norm | 0.5 | Maximum norm for gradient clipping |
| num_envs | 512 | Number of parallel environments |
| num_steps | 128 | Number of steps per policy rollout |
| total_timesteps | 20000000 | Total number of training timesteps |

Table 4: Hyperparameters for BlendRL training.

| Layer | Configuration |
|---|---|
| Fully Connected Layer 1 | Linear($N_{in}$, 120) |
| Activation 1 | ReLU() |
| Fully Connected Layer 2 | Linear(120, 60) |
| Activation 2 | ReLU() |
| Fully Connected Layer 3 | Linear(60, $N_{out}$) |

Table 5: Object-centric critic networks for BlendRL.

---

**Algorithm 1** BlendRL Policy Reasoning

---

**Input:** $\pi_\theta^{neural}, \pi_\phi^{logic}, V_\mu^{CNN}, V_\omega^{OC}$, blending function $B$, state $(\boldsymbol{x}, \boldsymbol{z})$
 1: $\beta = B(\boldsymbol{x}, \boldsymbol{z})$ # Compute the blending weight $\beta$
 2: $action \sim \beta \cdot \pi_\theta^{neural}(\boldsymbol{x}) + (1 - \beta) \cdot \pi_\phi^{logic}(\boldsymbol{z})$ # Action is sampled from the mixed policy
 3: $value = \beta \cdot V_\mu^{CNN}(\boldsymbol{x}) + (1 - \beta) \cdot V_\omega^{OC}(\boldsymbol{z})$ # Compute the state value using $\beta$
 4: **return** $action, value$

---

**NUDGE.** We used a public code[7] to perform experiments with the CleanRL training script. All hyperparameters are shared with the BlendRL agents, as described in Table 4. We used the same ruleset as BlendRL agents for NUDGE agents. The critic network on object-centric states is described in Table 5.

**Neural PPO.** We used an implementation of the neural ppo algorithm [8] from the CleanRL project. The agent consists of an actor network and a critic network, sharing their weights except for the last layer. The base network shared by the actor and the critic is shown in Table. 6. A linear layer with non-shared weights follows for each actor and critic after on top of the base network.

| Layer | Configuration |
|---|---|
| Convolutional Layer 1 | Conv2d(4, 32, 8, stride=4) |
| Activation 1 | ReLU() |
| Convolutional Layer 2 | Conv2d(32, 64, 4, stride=2) |
| Activation 2 | ReLU() |
| Convolutional Layer 3 | Conv2d(64, 64, 3, stride=1) |
| Activation 3 | ReLU() |
| Flatten Layer | Flatten() |
| Fully Connected Layer | Linear(64 * 7 * 7, 512) |
| Activation 4 | ReLU() |

Table 6: Layer configuration of the neural ppo agent.

### A.9 ENVIRONMENT DETAILS

We hereby provide details of the environments we used in our experiments. We used HackAtari[9], a framework that offers modifications of Atari environments to simply them or change them for the robustness test. In our experiments, the modifications we used are shown in Table 7.

| Environment | Option | Explanation |
|---|---|---|
| Kangaroo | disable_falling_coconut | Disable the falling coconut |
| | change_level0 | The first stage is repeated |
| | random_position | Randomize the starting position |
| Seaquest | No option | |
| DonkeyKong | change_level0 | The first stage is repeated |
| | random_position | Randomize the starting position |

Table 7: Options and explanations for different environments

---

[7]https://github.com/k4ntz/NUDGE

[8]https://github.com/vwxyzjn/cleanrl/blob/master/cleanrl/ppo_atari.py

[9]https://github.com/k4ntz/HackAtari

## A.10 Ablation study: neural vs. logic blending module

We here provide an ablation study on the logic blending module, reruning BlendRL agents using a neural one on *Kangaroo* and *Seaquest*. As shown in Table 8, the agents that encompass logic-based blending modules outperform the neural ones.

Figure 10 presents the entropies of the blending weights. An entropy value of 1.0 signifies that neural and logic policies are equally prioritized (with each receiving a weight of 0.5), while an entropy value of 0.0 indicates that only one policy is active, with the other being completely inactive. In both environments, the logic blending module consistently produced higher entropy values for the blending weights, indicating effective utilization of both neural and symbolic policies without overfitting to either one.

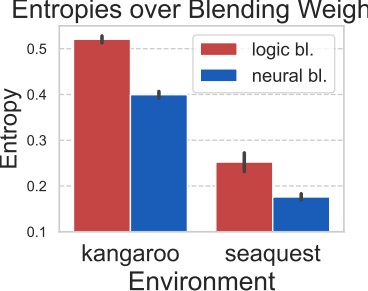

| Episodic Return | Kangaroo | Seaquest |
|---|---|---|
| Neural Blending | $91.6 \pm 43.6$ | $17.8 \pm 0.55$ |
| Logic Blending | $\mathbf{186}_{\pm\mathbf{12}}$ | $\mathbf{39.9}_{\pm\mathbf{7.12}}$ |

Table 8: **Comparison: Neural v.s. Logic for policy blending.** Average returns over 10 different random seeds are shown of trained agents. The logic blending module outperforms neural one consistently.

Figure 10: **Logic blending can keep both policies effective.** Entropies over blending weights are shown.

## A.11 Ablation study: Noise on Object-Centric States

We conducted additional experiments by introducing noise to the input object-centric states (only at test time). Specifically, we made some objects in the input state invisible at varying noise levels, ranging from 0.1 to 0.5. For example, a noise rate of 0.1 indicates that detected objects are invisible with a 10% probability. We evaluated 10 episodes and reported the mean episodic returns and lengths. Fig. 11 shows episodic returns and lengths for three environments. We observed the following facts: (i) Noise significantly impacts the overall performance of BlendRL agents. This is because the blending module relies on object-centric states, and the introduced noise can lead to incorrect decisions. For example, agents may mistakenly use logic policies when enemies are nearby. (ii) Noise affects episodic return more than episodic lengths overall. A notable exception is in the Seaquest environment, where episodic lengths decreased significantly due to noise. This suggests that high-quality object-centric states are crucial when reactive actions are frequently required, such as when there are many enemies. Obviously, training agents in noisy environments would increase the agents' robustness.

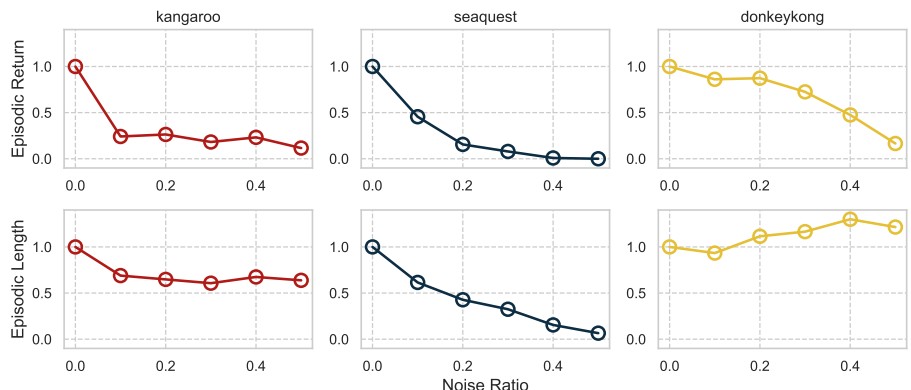

Figure 11: Episodic return and length with different noise ratios on object-centric states.

### A.12 PROGRESSIVE ENVIRONMENT ILLUSTRATION

As explained by Delfosse et al. (2024c), *Seaquest* is a progressive environment in which the agent first needs to master easy tasks, before being provided with more complex ones in newly unlocked parts of the environment, reflected by the number of enemy to be shot here.

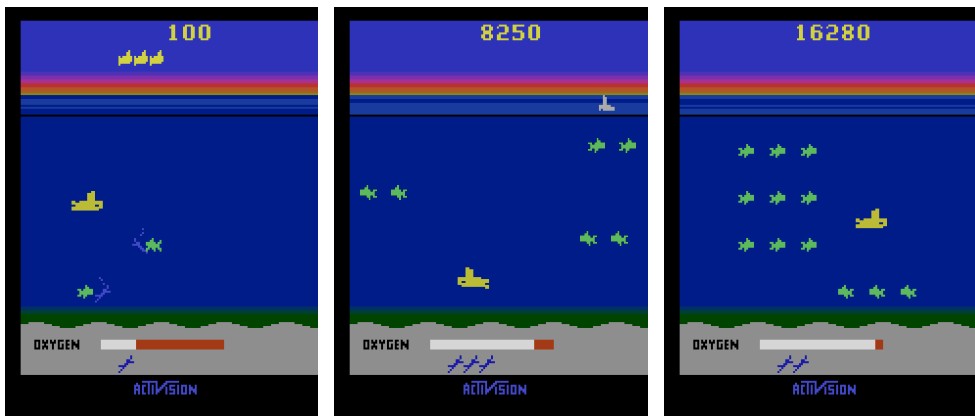

Figure 12: **Seaquest is a progressive environment.** The more an agent collects points/reward (depicted at the top), the more enemies are spawning

### A.13 DEITAILED BACKGROUND OF REINFORCEMENT LEARNING

We provide more detailed background for reinforcement learning. Policy-based methods directly optimize $\pi_\theta$ using the noisy return signal, leading to potentially unstable learning. Value-based methods learn to approximate the value functions $\hat{V}_\phi$ or $\hat{Q}_\phi$, and implicitly encode the policy, *e.g.* by selecting the actions with the highest Q-value with a high probability (Mnih et al., 2015). To reduce the variance of the estimated Q-value function, one can learn the advantage function $\hat{A}_\phi(s_t, a_t) = \hat{Q}_\phi(s_t, a_t) - \hat{V}_\phi(s_t)$. An estimate of the advantage function can be computed as $\hat{A}_\phi(s_t, a_t) = \sum_{i=0}^{k-1} \gamma^i r_{t+i} + \gamma^k \hat{V}_\phi(s_{t+k}) - \hat{V}_\phi(s_t)$ (Mnih et al., 2016). The Advantage Actor-critic (A2C) methods both encode the policy $\pi_\theta$ (*i.e.* actor) and the advantage

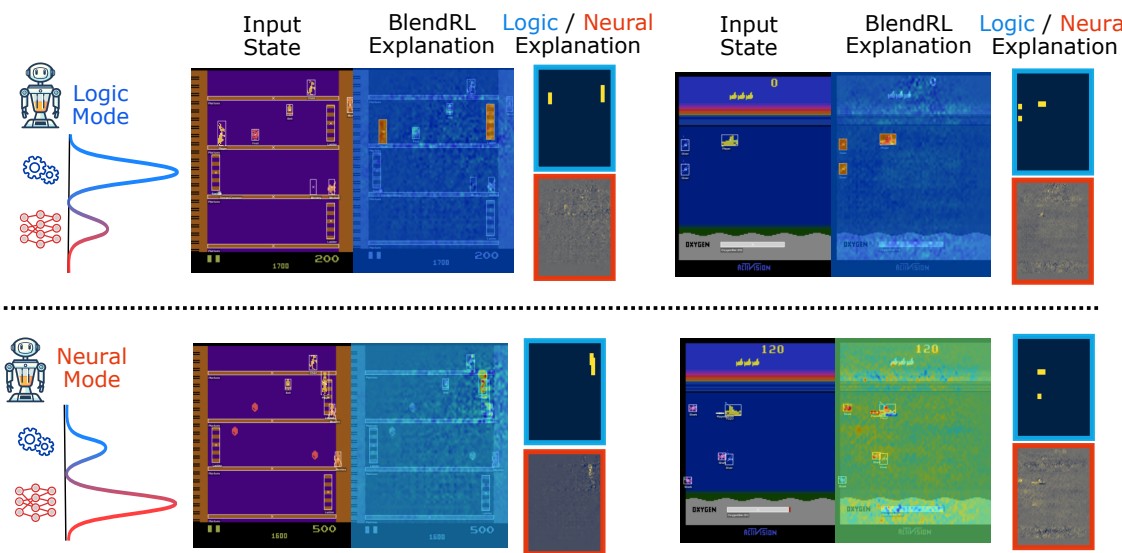

Figure 13: We depict BlendRL explanations on states where symbolic policy is prioritized (top) and those where neural policy is activated (bottom) on Kangaroo and Seaquest.

function $\hat{A}_\phi$ (*i.e.* critic), and use the critic to provide feedback to the actor, as in (Konda & Tsitsiklis, 1999). To push $\pi_\theta$ to take actions that lead to higher returns, gradient ascent can be applied to $L^{PG}(\theta) = \hat{\mathbb{E}}[\log \pi_\theta(a \mid s)\hat{A}_\phi]$. Proximal Policy Optimization (PPO) algorithms ensure minor policy updates that avoid catastrophic drops (Schulman et al., 2017), and can be applied to actor-critic methods. To do so, the main objective constraints the policy ratio $r(\theta) = \frac{\pi_\theta(a|s)}{\pi_{\theta_{old}}(a|s)}$, following $L^{PR}(\theta) = \hat{\mathbb{E}}[\min(r(\theta)\hat{A}_\phi, \text{clip}(r(\theta), 1 - \epsilon, 1 + \epsilon)\hat{A}_\phi)]$, where clip constrains the input within $[1 - \epsilon, 1 + \epsilon]$. PPO actor-critic algorithm's global objective is $L(\theta, \phi) = \hat{\mathbb{E}}[L^{PR}(\theta) - c_1 L^{VF}(\phi)]$, with $L^{VF}(\phi) = (\hat{V}_\phi(s_t) - V(s_t))^2$ being the value function loss. An entropy term can also be added to this objective to encourage exploration.

## A.14 BLENDRL EXPLANATION: A CASE STUDY

We conducted an additional ablation study examining how symbolic and neural explanations are blended over time. Fig. 13 showcases BlendRL's explanations of different states on Kangaroo and Seaquest. The top-row represents explanations when the logic policy is active, and the bottom-row represents those where neural policy is prioritized.

The observations are summarized as follows:

- When the symbolic policy is active, the logic policy provides clear explanations using relevant objects in an object-centric representation. The neural policy, however, produces very noisy explanations that neither highlight important objects nor illustrate the reasoning behind them.

- When the neural policy is active, the logic policy continues to produce solid explanations of objects but may miss critical ones necessary for effective action, such as an enemy (monkey) close to the agent. The neural policy, despite being noisy, provides informative explanations by highlighting relevant objects, indicating its ability to capture objects vital for reactive actions.

These observations suggest that the symbolic policy complements the neural policy, allowing the latter to focus more effectively on reactive actions.

## A.15  DETAILED EXPLANATIONS BY LLMS

Inspired by (Luo et al., 2024), we utilized LLMs to generate interpretable textual explanations on top of BlendRL's logical explanations. We instruct LLMs by specifying the task of explanation generation, providing an environment description, observing facts (BlendRL's logic explanation), and an action taken. We show the generated explanations for each scene in Fig. 5.

Kangaroo: *"The agent observed that it was positioned to the left of ladder1, which is crucial for ascending to the upper floors where the joey might be located. Since the player and ladder1 are on the same floor, the agent decided to move right towards ladder1. This action aligns with the task objective of reaching the joey, as navigating to the ladder is a necessary step to climb and advance to higher levels in the environment."*

Seaquest: *"The agent moved downwards because the player is positioned above a diver and there are no immediate threats present around the player. Since the divers have not been fully collected yet, the agent prioritized descending to rescue the diver below. This decision aligns with the task of rescuing divers while managing the absence of immediate external threats."*

DonkeyKong: *"The agent observed that the player was on the right side of ladder9 and on the same floor. To progress towards reaching the top and rescuing Princess Peach, the player needed to approach ladder9 to climb it and advance to a higher level. Therefore, the action taken was moving left, which allowed the player to position themselves directly over ladder9, enabling them to climb it and continue their ascent while avoiding obstacles."*

The generated textual explanations provide clearer insights into the action-making of the agents, enhancing their transparency.

The full prompts we used to generate the textual explanation are the following:

```
% Kangaroo
Generate an explanation of the action taken by the agent given task description , observed facts, and
    taken action. Show only the generated explanation. Be concise.

Task Description: In the Kangaroo environment, the task is to go to reach the joey. You will find
    monkeys and need tododge or fire them when they get close by. To reach the joey, the player needs
     to climb laddersto go to the upper floors.

Observed Facts: player is on left of ladder1, player and ladder1 are on the same floor
Taken Action: right

Textual Explanation:

% Seaquest
Generate an explanation of the action taken by the agent given task description , observed facts, and
    taken action. Show only the generated explanation.

In the Seaquest environment, the task is to rescue divers while managing oxygen levels. You will find
    enemy submarines and need to shoot or dodge them when they get close by. If divers are fully
    collected (6 divers), then go up to the surface to rescue them. Additionally, the player must
    monitor oxygen levels and ascend to the surface to replenish when low.

Observed Facts: player is above diver, nothing is aorund player, divers are not fully collected
Taken Action: down

Textual Explanation:

% DonkeyKong
```

```
Generate an explanation of the action taken by the agent given task description , observed facts, and
    taken action. Show only the generated explanation.

In the Donkey Kong environment, the task is to reach the top to rescue Princess Peach while avoiding
    obstacles. You will find barrels rolling towards the player and need to jump over or dodge them
    when they get close by. If the player reaches a ladder, they should climb it to advance to higher
     levels. Additionally, the player must monitor and avoid obstacles like flames that move across
    platforms.

Observed Facts: player is on right of ladder9, player is on the same floor as ladder9
Taken Action: left

Textual Explanation:
```

### A.16    ACTION DISTRIBUTION IN NEURAL AND LOGIC POLICIES

Figure 14 illustrates the distribution of actions taken by the logic and neural policies in a trained BlendRL agent, based on 10k steps in the Seaquest environment. While the logic components primarily execute reactive actions (e.g., *fire*), the neural agents predominantly handle complementary actions.

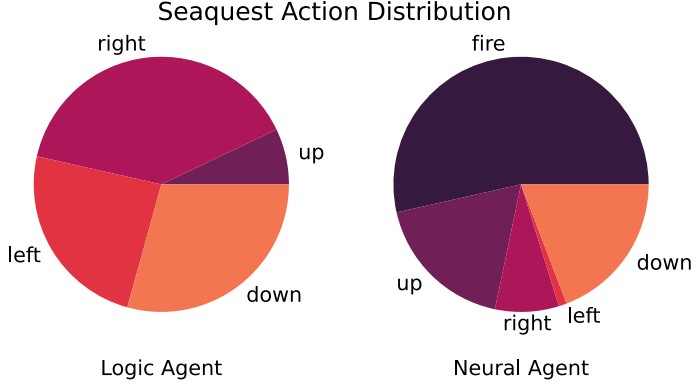

Figure 14: Action distribution of a BlendRL agent in Seaquest. We depict the proportion for each logic and neural agent, respectively.

### A.17    ABLATION: INDUCTIVE BIAS FOR LLM RULE GENERATION

To identify the significance of the few-shot examples in the rule-generation step, we provide an ablation study using imperfect rule examples for LLMs. Instead of using optimal rules obtained by NUDGE (Delfosse et al., 2023a), we provide non-optimal but grammatically correct rules. We used the following suboptimal rules:

```
Go right if an enemy is getting close by the player.
right(X):-closeby(Player,Enemy).

Jump to get the key if the player has the key and the player is on the right of the key.
jump_get_key(X):-has_key(Player),on_right(Player,Key).

Go right to go to the door if the player is close by an enemy and the player has the key.
right_go_to_door(X):-closeby(Player,Enemy),has_key(Player).
```

We compare the rule-generation performance with (i) 5 optimal rule examples from NUDGE policies, (ii) 3 suboptimal rule examples, (iii) 1 suboptimal rule example[10], and (iv) no rule example.

---

[10]We used the first example with `right` and `closeby`.

Table 9 shows the success rate of the rule generation on 10 trials on GPT-4o. If a generated rule set represents the same policy as those in Fig. 4 and Section A.5 with the original prompt, we count it as a success. For all three environments, GPT4-o generated correct action rules with 3 suboptimal rule examples. The performance decreased by reducing the number of examples, especially in Seaquest. Without examples, it failed to generate valid action rules.

| Methods | Kangaroo | Seaquest | DonkeyKong |
|---|---|---|---|
| NUDGE Rules (5 Examples) | 100% | 100% | 100% |
| Suboptimal Rules (3 Examples) | 100% | 100% | 100% |
| Suboptimal Rule (1 Example) | 90% | 10% | 90% |
| No Examples | 0% | 0% | 0% |

Table 9: Success rate of action rule generation with optimal NUDGE rules, sub-optimal rule examples, and without any rule example.

Without examples, incorrectly formatted rules are often generated, *e.g.*, for Seaquest:

```
% Failure cases in Seaquest
go_up_diver(X):-deeper(X,diver),visible(diver),not(full(divers)).
go_up_rescue(X):-collected(full(divers)).
go_left(X):-right_of(X,diver),visible(diver),not(collected(full(divers))).
go_right(X):-left_of(X,diver),visible(diver),not(collected(full(divers))).
go_up_diver(X):-deeper(X,diver),visible(diver),not(collected(full(divers))).
go_down_diver(X):-higher(X,diver),visible(diver),not(collected(full(divers))).
```

This result suggests that the provided few-shot rule examples inform LLMs about the general rule structure to define actions, but they do not convey the semantics or specific strategies required for the environments.

