# OpenReview forum: "BlendRL: A Framework for Merging Symbolic and Neural Policy Learning"
_ICLR.cc/2025/Conference — ICLR 2025 Spotlight_

### Official Review · Reviewer_pgu6 · 2024-10-17

**Soundness:** 3
**Presentation:** 4
**Contribution:** 3
**Rating:** 8
**Confidence:** 3

**Summary:**

The paper presents BlendRL, a neuro-symbolic reinforcement learning framework that integrates symbolic reasoning and neural policies, aiming to improve agent performance by mixing neural and symbolic policies. The authors demonstrate BlendRL’s efficacy in classic Atari games, highlighting that the framework outperforms purely neural baselines and neuro-symbolic baselines like NUDGE.

**Strengths:**

The paper is clearly written, and the proposed framework is compelling in its attempt to bridge the gap between symbolic reasoning and neural policy learning. The dual use of symbolic logic for high-level reasoning and deep neural networks for low-level reaction is well-motivated and timely. The experiments convincingly show that BlendRL agents perform better than existing methods, particularly in environments that demand both reasoning and quick reflexes. For instance, in Seaquest, the neural policy was able to effectively manage shooting enemies, while the symbolic component handled pathfinding and resource collection, demonstrating how the hybrid model synergistically enhances performance.

**Weaknesses:**

Overall, I think this work will have a positive impact on the community, but I still have some concerns:
1. The paper evaluates the proposed method in a limited set of experimental environments. It would strengthen the validation of the method if the authors could evaluate BlendRL in a wider variety of environments, especially those with different characteristics (such as the environments BeamRiderNoFrameskip-v4, EnduroNoFrameskip-v4, and QbertNoFrameskip-v4).
2. While the paper demonstrates strong performance, it would be more impactful if the method were tested in more realistic environments such as IsaacGym[1] or MetaDrive[2]. These environments pose more challenging and realistic scenarios, which could further validate the generalizability and robustness of BlendRL. It would also be valuable for the authors to discuss any challenges or adjustments required to apply BlendRL to these more complex environments, and to comment on how they expect its performance and benefits to scale in such domains.
3. The explanation of how symbolic policies contribute to the overall behavior of mixed policies could benefit from more clarity. It would be helpful if the authors provided a detailed analysis or specific examples showing how the symbolic explanations correspond to the mixed policy's actions. Including case studies that illustrate how explanations evolve as the blending between neural and symbolic components changes could enhance understanding of the interpretability and functionality of the method.

[1] Makoviychuk, Viktor, et al. "Isaac gym: High performance gpu-based physics simulation for robot learning." arXiv preprint arXiv:2108.10470 (2021).

[2] Li, Quanyi, et al. "Metadrive: Composing diverse driving scenarios for generalizable reinforcement learning." IEEE transactions on pattern analysis and machine intelligence 45.3 (2022): 3461-3475.

**Questions:**

1. Figure 6 references textual interpretation; could you kindly provide a more detailed explanation?
2. In cases where the logic chain becomes quite long, would the policy still be considered explainable?

---

> ### Author Response · Authors · 2024-11-21
> **Response to Reviewer pgu6 (1/2)**
>
> We thank the reviewer for the thoughtful comment and for acknowledging that the paper is clearly written and the LLM utilization for logic policy generation is innovative.
>
> > experiments in a wider variety of environments (such as the environments BeamRider, Enduro, and Qbert)
>
> Thank you for your suggestion. We agree that more experiments would strengthen the result. Regarding the Atari environments, we are particularly working on integrating Qbert, as it also involves both high-level reasoning (counting and path planning) and reactive actions (dodge enemies). Unfortunately, the other 2 environments are not fully covered by OCAtari, and we are trying Luo, et al.'s extraction framework [1] as an alternative.
>
> > experiments in more realistic environments such as IsaacGym or MetaDrive.
>
> Thank you for your suggestion. Experiments on MetaDrive would be very promising, as agents are exposed to complex and real-time decision-making scenarios that differ from Atari ones. However, we will leave it for future work, as the current experiments are already addressing much more complex environments compared to those used in the logic-based RL agents upon which BlendRL is built. Further, these environments already support our claim that **neural equipped symbolic agents can overcome insufficient prior knowledge.** For more details, please refer to the general remark.
>
> > It would be helpful if the authors provided a detailed analysis or specific examples showing how the symbolic explanations correspond to the mixed policy's actions.
>
> To address this, we conducted an additional ablation study examining how symbolic and neural explanations are blended over time.  Here is a summary of our observations:
>
> - When the symbolic policy is active:
>   - The transparent logic policy provides clear explanations through the interpretable rules and the object-centric representations.
>   - The neural policy, however, produces noisy explanations that cannot illustrate the reasoning behind them.
>
> - When the neural policy is active:
>   - The logic policy continues to make use of transparent rule but may miss critical concepts necessary to produce the optimal behavior, such as the threatening enemies (*e.g.* monkeys in *Kangaroo*, sharks in *Seaquest*, barrels in *DonkeyKong*.
>   - The neural policy, despite being noisy, provides informative explanations by highlighting relevant objects, indicating its ability to capture objects vital for reactive actions. The inclusion of INSIGHT's distillation of the neural policy into EQL, together with the LLM's explanations, can crucially benefit BlendRL agents, and allow them to have fully interpretable behavior when using both systems. We detail this within the new version of the manuscript, as we believe that this constitutes the most interesting line of direction for future work.
>
> These observations suggest that *the symbolic policy complements the neural policy, allowing the latter to focus more effectively on reactive actions.*
>
> **We have included these results and discussions in the revised manuscript (pp.28-29, lines 1305-1322).**  To this end, we also curated a working demonstration on how BlendRL’s explanations evolve over time. Please refer to [the anonymous link](https://anonymous.4open.science/r/anon-blendrl-BA06/explanations/README.md).
>
> Thank you for your suggestion, which has enhanced the clarity of the mechanism behind the proposed blended policies.
> If you have any suggestions on this part, we would gladly improve it.

---

> ### Author Response · Authors · 2024-11-21
> **Response to Reviewer pgu6 (2/2)**
>
> > Figure 6 references textual interpretation; could you kindly provide a more detailed explanation?
>
> This is a good point. **Inspired by INSIGHT [1], we elaborated the explanation by using LLMs dedicated to explanations of actions.** To achieve this, we provided a textual prompt to generate a detailed and interpretable textual explanation from the task description, observed facts (BlendRL’s logic explanation), and taken action.
>
> We show the generated explanations for each scene in Fig. 6:
>
> Kangaroo:
> *”The agent observed that it was positioned to the left of ladder1, which is crucial for ascending to the upper floors where the joey might be located. Since the player and ladder1 are on the same floor, the agent decided to move right towards ladder1. This action aligns with the task objective of reaching the joey, as navigating to the ladder is a necessary step to climb and advance to higher levels in the environment.”*
>
> Seaquest:
> *”The agent moved downwards because the player is positioned above a diver and there are no immediate threats present around the player. Since the divers have not been fully collected yet, the agent prioritized descending to rescue the diver below. This decision aligns with the task of rescuing divers while managing the absence of immediate external threats.”*
>
> DonkeyKong:
> *”The agent observed that the player was on the right side of ladder9 and on the same floor. To progress towards reaching the top and rescuing Princess Peach, the player needed to approach ladder9 to climb it and advance to a higher level. Therefore, the action taken was moving left, which allowed the player to position themselves directly over ladder9, enabling them to climb it and continue their ascent while avoiding obstacles.”*
>
>
> The generated textual explanations provide clearer insights into the action-making of the agents, enhancing their transparency.  **We have included these results with full prompts we used in the revised manuscript (pp. 29-30, lines1324-1376, Section A.15), and added a reference in the caption of Fig.6 (pp. 8, line 340).** Thank you for your suggestion.
>
>
> > In cases where the logic chain becomes quite long, would the policy still be considered explainable?
>
> Yes, the BlendRL policy is still explainable for long-chained reasoning, as it explicitly encodes the forward-reasoning process in FOL, providing formal traces of proof for many multiple steps of reasoning. To provide more insights on this, **we added 3-page of a dedicated explanation for forward reasoning with a working example from Kangaroo (pp. 16-18, Section A.1).**
>
> However, when relying on too many rules, any transparent symbolic policy becomes non-interpretable. To resolve that, one can use insight from Luo's et al. work [1] again, *i.e.* ask an LLM to *concisely* explain the logic based policy. Otherwise, in line with Kohler's et al. [2], we could try to focus on producing concise policies (*i.e.* limit the set of potential rules produced).
>
> **We added this discussion to the manuscript (pp.10, lines 425-432).**
>
>
>
> Thank you once again for your valuable comments and insightful questions. We believe the manuscript has been significantly improved by incorporating your feedback. We hope we have addressed them adequately in the revised manuscript. We are happy to answer any further questions you may have.
>
>
>
>
> -------
> [1] Luo, et al. (2024) End-to-End Neuro-Symbolic Reinforcement Learning with Textual Explanations. ICML.
>
> [2] Kohler, et al. (2024)  "Interpretable and Editable Programmatic Tree Policies for Reinforcement Learning." Workshop on Interpretable Policies in Reinforcement Learning@ RLC.

---

> > ### Author Response · Authors · 2024-11-25
> > **Any further questions?**
> >
> > Dear Reviewer,
> >
> > Since the discussion phase is coming to a close, we would like to ask if there are any outstanding concerns. We have provided detailed comments to your concerns and hope that we have cleared all misunderstandings.
> >
> > If yes, it will be great if you could reconsider your score.
> >
> > Regards,
> >
> > Authors

---

> ### Comment · Reviewer_pgu6 · 2024-11-25
>
> I appreciate the authors' thorough and thoughtful response. The expanded discussions and new experimental results in the updated version have strengthened my confidence in the value of this work. I am revising my score to 8 as a result."

---

### Official Review · Reviewer_NRGe · 2024-10-31

**Soundness:** 3
**Presentation:** 3
**Contribution:** 3
**Rating:** 8
**Confidence:** 4

**Summary:**

The paper proposed to jointly learn a mixture of neural policy and symbolic policy (represented in a differentiable way) via reinforcement learning. In the experiments, the neural policy is simply a CNN-based policy, the differentiable symbolic policy is based on a method called differentiable forward reasoner, and the weights of the mixture can themselves be seen as the output of a trainable policy. The proposed method is tested on three Atari games.

**Strengths:**

The idea of learning a mixture of neural policy and symbolic policy is novel. Given the presented experimental results, it seems to lead to nicely interpretable decision rules (at least in the symbolic part). Also, surprisingly to me, it seems that it automatically learns to make "reflex" decisions preferably using the neural policy.

The overall proposition nicely integrates several previous techniques to offer an end-to-end method, with little human inputs.

**Weaknesses:**

The paper should be more self-contained. For instance, this work builds on several existing propositions (notably by Shindo et al. and Delfosse et al.). The authors should recall and provide more details of those previous works to help the reader more easily understand BlendRL and appreciate its novelty. For instance, I think the current presentation of differentable forward reasoner is too light, e.g., how is the set \mathcal C related to the forward reasoning graph? What's the advantages/disadvantages of this representation using a graph compared to other frameworks, e.g., dILP?
Also, Section 3.3,  which seems to me to be a key component that can have a huge impact on BlendRL, is lacking in details. For instance, I suggest the authors to provide more information about step (i). In addition, doesn't using the NUDGE policies to generate examples for the LLM provide a strong inductive bias to BlendRL?

The experimental validation is missing some details and maybe a bit light. The authors should explain how they chose their hyperparameters (e.g., Entropy coefficient for blending regularization). Since BlendRL requires a more complex policy model than Neural PPO or NUDGE), I believe it's possible via (potentially costly) hyperparameter tuning to obtain better results for BlendRL.
The authors only evaluates on three Atari games, which are moreover different from those used in NUDGE. Why is that so? Would BlendRL still outperform NUDGE on the tasks where it has demonstrated a good performance?

Although I haven't followed closely the latest developments in neuro-symbolic approaches, I believe that the literature in this direction is quite rich. For instance, I think that there are other works trying to combine System 1 and System 2 capabilities in AI agents and there are other neuro-symbolic RL approaches. The paper could be improved by better situating BlendRL in the existing literature.

Minor points:
There are a number of typos that should be corrected, e.g.,
- line 107: I believe that function symbols are not used in this work. This should be clarified.
- line 162: What's called "state" actually corresponds to "observation"
- line 177: Regarding the dimension of the logic policy, should F appear?
- lines 852-855: the learning rates are missing?

**Questions:**

1) If I wanted to apply it to a new game, what should be done to generate the action rules (Section 3.3)?

2) Does using the NUDGE policies to generate examples for the LLM provide a strong inductive bias to BlendRL?

3) How were the hyperparameters obtained?

4) How did you choose the three Atari games? Why did you use different ones compared to those used in NUDGE?

---

> ### Author Response · Authors · 2024-11-21
> **Response to Reviewer NRGe (1/3)**
>
> We thank the reviewer for their insightful comment and for recognizing that the proposed BlendRL framework offers enhanced interpretability, as well as for acknowledging the surprising empirical result of the successful integration of neural and logic policies.
> We hereafter address the reviewer's concerns.
>
> >  The paper should be more self-contained. … The authors should recall and provide more details of those previous works.
>
> As suggested by the reviewer, we now provide detailed explanations of BlendRL's background. **The revised version contains a dedicated 3-page section (pp. 16-19 Section A.1) for (differentiable) forward reasoning** with the following subsections:
>
> - What is forward reasoning?:
> - What is differentiable forward reasoning?
> - Why graph-based reasoning?:
>
> Moreover, we added a consistent working example on *Kangaroo* throughout the explanation with elaborated figures. This provides an intuitive understanding and makes the paper self-contained.
>
>
> Let us now provide concise answers to each specific questions.
>
> > how is the set $\mathcal{C}$ related to the forward reasoning graph?
>
>
> The set of clauses $\mathcal{C}$ defines the structure of the forward reasoning graph.
> Essentially, given $\mathcal{C}$, BlendRL encodes them by grounding them (removing variables). A working example of this encoding process is available in the revised section (A.1 on pages 16-19).
>
> > What's the advantages/disadvantages of this representation using a graph compared to other frameworks, e.g., dILP?
>
> The advantage of using a graph is its memory efficiency.
> The dILP approach, which primarily uses tensors, consumes memory quadratically with the number of (ground) rules and (ground) facts.
> This severely limits its applicability for training on parallelized (vectorized) environments, which is crucial for successful training of RL agents. Graph-based encoding overcomes this bottleneck, and thus, we employ it for BlendRL.
>
> We empirically observed that memory efficiency is the key to BlendRL agents' successful training. In *Seaquest*, the tensor-based one scaled up to most 100 environments on NVIDIA A100-SXM4-40GB GPU.  However, the graph-based one scaled to more than 500 parallelized environments, significantly improving performance.
>
> **We added this discussion to the revised version (pp. 19, lines 856-862) in Section A.1.**
>
> > Also, Section 3.3, which seems to me to be a key component that can have a huge impact on BlendRL, is lacking in details. For instance, I suggest the authors to provide more information about step (i).
>
> To address this, **we added a dedicated 1-page section (pp. 20-21, Section A.3)** in the appendix. Let us explain it briefly. We follow 3 steps;
>
> (Step 1) We provided general task instructions of the predicate generation task as a prompt.
>
> (Step 2) Subsequently, we provide examples demonstrating predicate generation. All the environments used here (GetOut, 3Fish, and Heist) are adapted from the public repository of NUDGE, which are non-Atari environments. We have added textual task descriptions for each environment, pairing them with the corresponding output.
>
> (Step 3) Finally, we offer environment-specific instructions, detailing the task descriptions for each task in interest, i.e., *Kangaroo*, *Seaquest*, and *DonkeyKong*.
>
> **The full prompts are available in the revised section (pp. 20-21, Section A.3).**
>
>
> > In addition, doesn't using the NUDGE policies to generate examples for the LLM provide a strong inductive bias to BlendRL?
>
> The examples provided to the LLM are expert-provided rules used by NUDGE agents in non-atari environments. However, we believe that the LLM would incorporate the knowledge necessary to produce the rules, and that the examples allows it to provide a grammatically correct rule set.
>
> > If I wanted to apply it to a new game, what should be done to generate the action rules (Section 3.3)?
>
> A task description of the game should be provided in natural language text. **We added Section A.2 (pp. 19-20, lines 868-925) that describes the full prompts for rule generation** used in the experiments. Please also refer to our reply to Reviewer ***NFD8***.
>
> > How were the hyperparameters obtained? (e.g., Entropy coefficient for blending regularization).
>
> Most of the hyperparameters are the default values in the [CleanRL learning script](https://github.com/vwxyzjn/cleanrl/blob/master/cleanrl/ppo_atari.py).
> For newly introduced parameters (*e.g.* the entropy coefficient), we have tried promising values that can be estimated from other regularization terms and then performed 2 or 3-step tuning.

---

> ### Author Response · Authors · 2024-11-21
> **Response to Reviewer NRGe (2/3)**
>
> > How did you choose the three Atari games? Why did you use different ones compared to those used in NUDGE?
>
> Atari is one of the most popular environments for evaluating RL agents. We selected environments that inherently include multiple goals (avoiding/destroying obstacles and reach desired goal (as well as maintaining oxygen for *Seaquest*), require both abstract reasoning and reactive actions. In contrast, the environments used in NUDGE are synthetic, contain a single goal, and focus mainly on abstract reasoning without involving reactive actions. Therefore, the selected Atari environments are more challenging and accurate to demonstrate BlendRL agents' ability to learn beyond logic. *For more detailed discussions, please refer to the general remarks.*
>
>
>
> > Although I haven't followed closely the latest developments in neuro-symbolic approaches, …  the paper could be improved by better situating BlendRL in the existing literature.
>
> Firstly, we would like to clarify that in the related work section (Section 5, lines 415-434), we revisit significant literature on neuro-symbolic approaches in reinforcement learning (RL). Following the reviewer's suggestion, **we also conducted an additional survey on the specific topic of *fast and slow thinking in RL*.**
>
> Botvinick et al. [1] propose a new RL paradigm with a nested learning-loop structure, consisting of an inner loop and an outer loop. The outer loop broadly explores parameters, aiding the inner loop in quickly adapting to specific tasks. RL$^2$ [2] introduces "slow" learning into RL agents by encoding the algorithm using a recurrent neural network (RNN), which is trained via a general-purpose RL algorithm. Tan et al. [3] implement slow thinking as a neural network with access to memory for storing and retrieving additional information during inference. Anthony et al. [4] achieve slow thinking by integrating planners and neural networks, enabling them to generalize generated plans.
>
> Collectively, these works offer diverse interpretations of fast and slow systems through learning loops [1], recurrent neural architectures [2], external memory access [3], and tree search [4].
>
> Our paper presents a novel approach that directly integrates differentiable logical reasoning into the policy rather than as an external module. This integration allows BlendRL agents to be trained end-to-end using established RL algorithms (e.g., A2C) and provides insights to the community for developing better-integrated System 1 and System 2 in RL.  **We added the discussions to the revised related work section (pp.10, lines 444-448), highlighted in blue.**
>
> -------
>
>
> [1] Botvinick et al.: Reinforcement Learning, Fast and Slow, Trends in Cognitive Sciences, 2019
>
> [2] Duan et al.: RL^2: Fast Reinforcement Learning via Slow Reinforcement Learning, ICLR 2017
>
> [3]  Tan et al.: Learning, Fast and Slow: A Goal-Directed Memory-Based Approach for Dynamic Environments, ICDL 2023
>
> [4] Anthony et al.: Thinking Fast and Slow with Deep Learning and Tree Search, NeurIPS 2017
>
> -------
>
> To this end, in response to Reviewer ***xzvw***'s suggestion, we conducted an expanded survey on neuro-symbolic approaches.  Please refer to our response to Reviewer ***xzvw*** for more details.
>
>
> We believe that we sufficiently positioned our work within the existing literature, as recommended by the reviewer.

---

> ### Author Response · Authors · 2024-11-21
> **Response to Reviewer NRGe (3/3)**
>
> Now, let us address the minor points.
>
> > line 107: I believe that function symbols are not used in this work. This should be clarified.
>
> **We have added a clarification as suggested by the reviewer (pp. 3, lines 103 and 139-140 (footnote)).** Let us elaborate on the use of function symbols in BlendRL.
>
> Graph-based reasoning systems can manage complex programs with function symbols, effectively overcoming the memory bottleneck [1]. Consequently, BlendRL is also capable of handling them, although our current experiments do not require their use. Incorporating programs with structured terms using function symbols, such as lists and trees, would be a significant enhancement, as seen in [meta interpreters](https://www.metalevel.at/acomip/). We intend to explore the integration of these advanced programs in future research, as the current paper focuses on the fundamentals of integrating neural and symbolic policies.
>
> **We have included this discussion in the revised manuscript (pp. 19, lines 863-868).** Thank you for highlighting this point.
>
> [1] Shindo et al.: Learning Differentiable Logic Programs for Abstract Visual Reasoning, Machine Learning, 2024.
>
> > line 162: What's called "state" actually corresponds to "observation"
>
> "Observation" could mislead readers into thinking of (incomplete) observations (i.e. subset of the states) of POMDP environments, but we rewrote "observable screen" for clarity.
>
> > line 177: Regarding the dimension of the logic policy, should F appear?
>
> No, due to the computational cost, BlendRL does not allow multiple frames as input for the logic policy at the current state. **We corrected this in the revised manuscript (pp.4, lines162-163).**
>
>
>
> > lines 852-855: the learning rates are missing?
>
>
> No, this table contains the learning rates for each trainable component.
> In the table *blender learning rate* represents the learning rate for the blending module, *learning rate* represents that for the neural policy, and *logic learning rate* represents that for the logic policy. **We added these clarifications in the caption of Table 4 (pp. 24, lines 1119-1121).**
>
>
> Thank you once again for your valuable comments and insightful questions. We believe the manuscript has been significantly improved by incorporating your feedback. As most of the concerns were related to presentation and clarification, we hope we have addressed them adequately in the revised manuscript. We are happy to answer any further questions you may have.

---

> > ### Author Response · Authors · 2024-11-25
> > **Any further questions?**
> >
> > Dear Reviewer,
> >
> > Since the discussion phase is coming to a close, we would like to ask if there are any outstanding concerns. We have provided detailed comments to your concerns and hope that we have cleared all misunderstandings.
> >
> > If yes, it will be great if you could reconsider your score.
> >
> > Regards,
> >
> > Authors

---

> > > ### Comment · Reviewer_NRGe · 2024-11-26
> > >
> > > I really appreciate that the authors have updated their paper with additional explanation to make the paper more self-contained and more accessible to the reader. Most of my concerns were addressed, except the point raised by my question 2:
> > >
> > > Wouldn't it be easy to check if the conjecture in your answer is correct by using a random policy described with a grammatically correct rule set instead of the good policy obtained with NUDGE?

---

> ### Author Response · Authors · 2024-11-26
> **Thank you for your response**
>
> We appreciate your insightful suggestion.
> To identify the significance of the few-shot rule examples in the rule-generation step, we provide an ablation study using imperfect rule examples for LLMs. As suggested by the reviewer, instead of using optimal rules obtained by NUDGE, we provide non-optimal but grammatically correct rule examples.
> We used the following suboptimal rules:
> ```
> Go right if an enemy is getting close by the player.
> right(X):-closeby(Player,Enemy).
>
> Jump to get the key if the player has the key and the player is on the right of the key.
> jump_get_key(X):-has_key(Player),on_right(Player,Key).
>
> Go right to go to the door if the player is close by an enemy and the player has the key.
> right_go_to_door(X):-closeby(Player,Enemy),has_key(Player).
> ```
>
> We compare the rule-generation performance with (i) 5 optimal rule examples from NUDGE policies (the original prompt), (ii) 3 suboptimal rule examples, (iii) 1 suboptimal rule example (the first example), and (iv) no rule example.
>
> The table below shows the success rate of the rule generation on 10 trials on GPT-4o. If a generated rule set represents the same policy as those in Fig. 4 and Section A.5 with the original prompt, we count it as a success.
> For all three environments, GPT4-o generated correct action rules with 3 suboptimal rule examples.
> The performance decreased by reducing the number of examples, especially in Seaquest.
> Without examples, it failed to generate valid action rules.
>
> | Methods                  | Kangaroo | Seaquest | DonkeyKong |
> |--------------------------|----------|----------|------------|
> | NUDGE Rules (5 Examples)    | 100%     | 100%     | 100%       |
> | Suboptimal Rules (3 Examples) | 100%     | 100%     | 100%       |
> | Suboptimal Rule (1 Example)   | 90%      | 10%      | 90%        |
> | No Examples              | 0%       | 0%       | 0%         |
>
> Without examples, incorrectly formatted rules are often generated, *e.g.*, for Seaquest:
>
> ```
> % Failure cases in Seaquest
> go_up_diver(X):-deeper(X,diver),visible(diver),not(full(divers)).
> go_up_rescue(X):-collected(full(divers)).
> go_left(X):-right_of(X,diver),visible(diver),not(collected(full(divers))).
> go_right(X):-left_of(X,diver),visible(diver),not(collected(full(divers))).
> go_up_diver(X):-deeper(X,diver),visible(diver),not(collected(full(divers))).
> go_down_diver(X):-higher(X,diver),visible(diver),not(collected(full(divers))).
> ```
> This result suggests that the provided few-shot rule examples inform LLMs about the general rule structure to define actions, but they do not convey the semantics or specific strategies required for the environments.
>
> We have included these results and discussions in the revised manuscript (pp. 31, lines 1446-1483, Section A.17).
>
> Thank you again for your valuable suggestion. We hope we have clarified your concerns in the revised manuscript. We are happy to answer any further questions you may have.

---

### Official Review · Reviewer_xzvw · 2024-11-03

**Soundness:** 2
**Presentation:** 3
**Contribution:** 3
**Rating:** 8
**Confidence:** 4

**Summary:**

The paper presents BlendRL, a neuro-symbolic RL framework that allows agents to blend resoning from pixel and object level representations. Unlike previous  neuro-symbolic RL approach BlendRL learns concurrently it's low and haigh level policies. Specifically, a blending module, informed by rules generated by a language model, dynamically determines the optimal mix of neural and symbolic policies based on the task context.

Authors test BlendRL in Seaquest and Kangaroo (from Atari), where agents must alternate between quick responses and logical planning, results show that BlendRL consistently outperforms purely neural or symbolic agents.

**Strengths:**

The paper is very well written and easy to follow. The proposed framework is quite novel to the best of my knowdledge, since the level 2 and 1 systems are usually much more separated than in BlendRL, and the results against vanilla PPO or a symbolic approach (NUDGE) are promising.

**Weaknesses:**

My biggest concern with this work is the lack of empirical comparison with any other neuro-symbolic baselines, e.g. [1-4]

Since BlendRL also weights heavily on object-based representations and relational learning it would have been good (although I don't consider this critical)  to include some contrast with deep learning approaches for such kind of learning, e.g. [1, 5-6].

While the point above will raise my confidence on the paper, I still thinkt hat the novelty of the approach and the results included outweight the weakpoints.

[1] Borja G. León, Murray Shanahan, and Francesco Belardinelli. In a nutshell, the human asked for this: Latent
goals for following temporal specifications. In International Conference on Learning Representations, 2022.

[2] Kuo, Yen-Ling, Boris Katz, and Andrei Barbu. "Encoding formulas as deep networks: Reinforcement learning for zero-shot execution of LTL formulas." 2020 IEEE/RSJ International Conference on Intelligent Robots and Systems (IROS). IEEE, 2020.

[3] Vaezipoor, Pashootan, et al. "Ltl2action: Generalizing ltl instructions for multi-task rl." International Conference on Machine Learning. PMLR, 2021.

[4] Qiu, Wenjie, Wensen Mao, and He Zhu. "Instructing goal-conditioned reinforcement learning agents with temporal logic objectives." Advances in Neural Information Processing Systems 36 (2024).

[5] Shanahan, Murray, et al. "An explicitly relational neural network architecture." International Conference on Machine Learning. PMLR, 2020.

[6] Feng, Fan, and Sara Magliacane. "Learning dynamic attribute-factored world models for efficient multi-object reinforcement learning." Advances in Neural Information Processing Systems 36 (2024).

**Questions:**

See weaknesses above

--- Post rebuttal ---

After reading Author's responses and the updated paper I have no more concerns. I believe this will be an interesting contribution for those working with neuro-symbolic approaches.

---

> ### Author Response · Authors · 2024-11-21
> **Response to Reviewer xzvw**
>
> We thank the reviewer for acknowledging that our paper is well written, and the proposed framework is novel in integrating neural and symbolic policies effectively.
> Let us address the raised issues.
>
> > My biggest concern with this work is the lack of empirical comparison with any other neuro-symbolic baselines, e.g. [1-4]
>
> Thank you for your suggestion.  We have added comparison to another deep baseline (namely DQN [R1]), as well as many symbolic baselines (NRLR [R2], SCoBots [R3], INTERPRETER[R4] and INSIGHT[R5]). While we are extending the experiment on INSIGHT, we already modified the manuscript to indeed compare with more baselines (cf Table 1, pp. 7).
>
> Let us clarify that the mentioned articles [1-4] all rely on Goal Conditionned RL (GCRL), which is not the focus of our work. In details:
>
> [1] proposes a new neural architecture to solve OOD instructions expressed in temporal logic. [2] proposes a framework to encode commands (instructions) written in linear temporal logic into deep RL agents. The agent takes a Linear Temporal Logic (LTL) formula as input and determines satisfying actions. [3] proposes LTL2Action, which teaches Deep RL agents to follow instructions in multi-task environments. LTL instructions are encoded through Relational GCN. [4] proposed a goal-conditioned RL framework to follow arbitrary LTL specifications. Contrary to previous approaches. It does not require sampling a large set of LTL instructions from the task space during training as goals.
>
> The critical difference is that, in BlendRL, logic expresses the policy itself and directly influences decision-making, and agents learn the underlying logic in environments. In contrast, the central focus of these studies [1-4] is to train deep neural networks using logical constraints (e.g., "get Gem" and "go Factory") provided as input, effectively guiding the neural agents.
> Consequently, none of them are evaluated Atari, because Atari environments usually do not provide sub-goals, e.g. *“satisfy C, A, B in that order, and satisfy D, A in that order”*, which are the main focus of these LTL-based studies. Thus, it is not trivial to incorporate them as a baseline for our evaluation setup.
>
> Moreover, we consider our contribution to be complementary to these works. Investigating the integration of LTL instructions within BlendRL promises to be particularly intriguing. This could pave the way for research where BlendRL's symbolic policies directly incorporate LTL instructions while neural policies are trained to adhere to them, emphasizing reactive actions. We intend to pursue this exploration in future work.
>
> **We added these discussions to the revised related work section (pp.10, lines 441-444, highlighted in blue).** Thank you for your suggestion.
>
>
> >  it would have been good (although I don't consider this critical) to include some contrast with deep learning approaches for such kind of learning, e.g. [1, 5-6].
>
> We appreciate your insight. Indeed, these works effectively incorporate object-centric representations and relational concepts to solve tasks in which objects and their relations are key. **We thus discuss them in our updated related work section (pp. 10, lines 447-449).**
>
>
> Thank you once again for your valuable comments and insightful suggestions. We believe the manuscript has been significantly improved by incorporating your feedback. We hope we have addressed them adequately in the revised manuscript. We are happy to answer any further questions you may have.
>
> -------
>
> [R1] Nair, et al. (2015). Massively parallel methods for deep reinforcement learning. ICML Workshop
>
> [R2] Jiang, Z., & Luo, S. (2019). Neural logic reinforcement learning. ICML.
>
> [R3]  Delfosse, et al. (2024). Interpretable concept bottlenecks to align reinforcement learning agents. NeurIPS.
>
> [R4] Kohler, et al. (2024)  "Interpretable and Editable Programmatic Tree Policies for Reinforcement Learning." Workshop on Interpretable Policies in Reinforcement Learning@ RLC.
>
> [R5] Luo, et al. (2024) End-to-End Neuro-Symbolic Reinforcement Learning with Textual Explanations. ICML.

---

> > ### Comment · Reviewer_xzvw · 2024-11-24
> > **Reponse to authors**
> >
> > I want to thank the authors for their detailed response. Including neuro-symbolic baslines and all the additional discussion and insights included in the updated version icreases my confidence on the contribution of this work. I am updating my score accordingly.
> >
> > Just a last comment for the authors, as I was reviewing the changes I noticed that you only noted that the LLM used was gpt4-0 without specifiying the exact version. It has been demonstrated that closed models change greatly their capabilities over months, sometimes for worse while being labelled the same. For the sake of reproducibility I encourage the authros to specify the release used in their experiments

---

> > > ### Author Response · Authors · 2024-11-24
> > > **Response**
> > >
> > > Dear Reviewer,
> > >
> > > Thank you your response and updating your confidence score. We are happy that we were able to address all your concerns. It will also be helpful if you could also reconsider your overall score in light of this.
> > >
> > > We will be happy to answer any further questions from your end in the rebuttal period. We will also specify the exact version in the manuscript as you suggested.
> > >
> > > Regards,
> > >
> > > Authors

---

### Official Review · Reviewer_NFD8 · 2024-11-06

**Soundness:** 1
**Presentation:** 2
**Contribution:** 1
**Rating:** 5
**Confidence:** 4

**Summary:**

This paper integrates condition-based logic decisions with neural network-based reinforcement learning policies through an LLM-based hybrid module to address the shortcomings of both approaches. It achieved better results in three Atari games compared to standard PPO and logic-based method.

**Strengths:**

1. The paper is clearly written and easy to follow.
2. The utilization of the language model shows a certain level of innovation.

**Weaknesses:**

1. The overall concept is not particularly novel, with numerous similar works, such as the well-known fast and slow systems, already existing.
2. The paper and appendix lack crucial details on how the LLM generates rules and calculates hybrid probabilities, and to what extent this is based on the content provided in the prompts. This is essential for determining whether the method can generalize to more diverse tasks.
3. The paper consistently emphasizes complex tasks, yet the experimental environment is not particularly sophisticated. Atari is a basic, outdated, and relatively simple benchmark that has largely been mastered. The three selected tasks are not among the recognized challenging ones in Atari (indeed, they are relatively simple). Truly difficult tasks, such as Montezuma’s Revenge, would be more appropriate as experimental environments.
4. The experiments should compare against more advanced RL algorithms. For example, Agent57 has already achieved a score of 30k on Kangaroo (while blend RL scores less than 20k) and 1000k on Sequest (while blend RL scores less than 5k). Therefore, the current experimental results do not demonstrate the superiority of the method.
5. Overall, the experiments lean towards simpler explorations and lack ablation studies that would reveal the characteristics of the method. For instance, I couldn't find which LLM was used, nor were there detailed experiments on the impact of object-centric representations on the method.

**Questions:**

see weakness

---

> ### Author Response · Authors · 2024-11-21
> **Response to Reviewer NFD8 (1/2)**
>
> We thank the reviewer for the thoughtful comment and for acknowledging that the paper is clearly written and the LLM utilization for logic policy generation is innovative.
> Let us now address the raised concerns.
>
> > "The overall concept is not particularly novel, with numerous similar works, such as the well-known fast and slow systems, already existing".
>
> We strongly disagree with this statement. **All other reviewers *xzvw*, *NRGe*, *pgu6*** underline BlendRL’s novelty in its integration of neural and symbolic policy reasoning and learning.
>
> Reviewer ***xzvw***: *”The proposed framework is quite novel to the best of my knowdledge, since the level 2 and 1 systems are usually much more separated than in BlendRL.”*
>
> Reviewer ***NRGe***: *”The idea of learning a mixture of neural policy and symbolic policy is novel. … Also, surprisingly to me, it seems that it automatically learns to make "reflex" decisions preferably using the neural policy.”*
>
> Reviewer ***pgu6***: *”the proposed framework is compelling in its attempt to bridge the gap between symbolic reasoning and neural policy learning.“*
>
> Moreover, we cite exactly the fast and slow systems (the famous dual-process theory **from psychology**, which explains how **humans** think and make decisions) *in the first paragraph of our introduction*, and explain our motivation to build a foundation of System 1 and System 2 (fast and slow) in RL agents. Reviewer ***pgu6*** agrees in this regard: *”The dual use of symbolic logic for high-level reasoning and deep neural networks for low-level reaction is well-motivated and timely.“*
>
> We believe that the reviewer’s statement does not hold (not novel, numerous similar works, already existing). We would appreciate any reference to a publication that implements this tight neuro-symbolic policy integration for RL, *i.e.*, a framework that allows agents to reason on both neural and symbolic policies, and learn jointly how to merge them.
>
> > The paper and appendix lack crucial details on how the LLM generates rules and calculates hybrid probabilities.
>
> Thank you for pointing this out. We have corrected this in the uploaded new version of the manuscript. Let us break down our changes:
>
> #### How does LLM generate rules?
> To address this issue, **we added Section A.3 (pp. 19-20) that explains the rule generation step with complete prompts.**  Let us explain how to generate rules.
>
> As illustrated in Figure 3, the input to the Large Language Model (LLM) consists of a textual policy description submitted by any user. Our method involves transforming these descriptions into rules that adhere to a specific format, making them usable by the reasoning modules within BlendRL.
>
> In brief, our LLM rule extraction approach consists of three steps:
>
> 1. We begin by providing a general format instruction as a prompt, which outlines the structure of the rules that define actions.
> 2. Next, we include examples to demonstrate rule generation. All rules presented here are derived from a trained NUDGE agent in the GetOut environment [2], a non-Atari environment.
> 3. Finally, we supply environment-specific instructions, detailing the complete set of action prompts for each environment.
>
> Please refer to the revised section (A.3, pp19-20) for the complete prompts.
>
> [2] Interpretable and explainable logical policies via neurally guided symbolic abstraction. NeurIPS 2023.
>
> Let us move on to the next point.
>
> #### How does the LLM compute the hybrid probabilities?
> The LLM does not compute the hybrid probabilities. The LLM is dedicated to rule generation.
> The probabilities are computed by a differentiable reasoner (blending function). We explicitly explain how to calculate each policy (pp. 4, lines 172--181) and merged action distributions (pp. 5, lines 232-247)  in Section 3.2.
>
> > The experiments should compare against more advanced RL algorithms. …  the current experimental results do not demonstrate the superiority of the method.
>
> Again, our goal is *NOT* to surpass all other RL algorithms or achieve state-of-the-art results on Atari environments, nor do we claim to do so. Instead, our contribution is orthogonal to the advanced RL algorithms mentioned, as BlendRL does not exclude neural or logic policies. In theory, we could integrate other frameworks, such as Agent57 (in place of PPO) to train these within BlendRL. However, without an official implementation available, we cannot reliably execute this integration.

---

> ### Author Response · Authors · 2024-11-21
> **Response to Reviewer NFD8 (2/2)**
>
> > Overall, the experiments lean towards simpler explorations… Truly difficult tasks, such as Montezuma’s Revenge, would be more appropriate as experimental environments.​​
>
> We disagree with the premise. We agree that *Montezuma’s Revenge* (or *Pitfall*) are notable for their difficulty, but mainly because of their sparse reward nature. The Atari environments used in our paper allow us to demonstrate that BlendRL agents can learn beyond logic, as even if not provided (*e.g.* by the LLM) with all the necessary concepts to solve the tasks, they can rely on their neural component, and solve the task. Please refer to the general remark for further details.
>
> > and lack ablation studies that would reveal the characteristics of the method.
>
>
> Thank you for your suggestion. Although we reported an ablation study in Section A.10 comparing symbolic and neural blending functions, we conducted an additional ablation study as suggested by the reviewer regarding the following concern:
>
> > nor were there detailed experiments on the impact of object-centric representations on the method.
>
> Logic policies process symbolic representations. We can thus not omit the object-centric (or symbolic) representations. However, we agree that more evaluations regarding the impact of quality of the object extraction will strengthen the paper.
> **We thus conducted additional experiments by introducing noise to the input object-centric states (only at test time).** Specifically, we made some objects in the input state invisible at varying noise levels, ranging from 0.1 to 0.5. For example, a noise rate of 0.1 indicates that detected objects are invisible with a 10% probability. We evaluated 10 episodes and reported the mean episodic returns and lengths. Fig, 11 (pp. 27, lines 1225-1239; also [Anonymous Link](https://anonymous.4open.science/r/anon-blendrl-BA06/assets/noise_ablation.pdf) is available) shows episodic returns and lengths for the three environments with different noise ratio.
> We observed the following facts:
> 1. Noise significantly impacts the overall performance of BlendRL agents. This is because the blending module relies on object-centric states, and the introduced noise can lead to incorrect decisions. For example, agents may mistakenly use logic policies when enemies are nearby.
> 2. Noise affects episodic return more than episodic lengths overall. A notable exception is in the Seaquest environment, where episodic lengths decreased significantly due to noise. This suggests that high-quality object-centric states are crucial when reactive actions are frequently required, such as when there are many enemies.
> Obviously, training agents on noisy environments, would increase the agents' robustness.
> **We added the new result and discussion in the revised manuscript (pp.26, lines 1211-1239; Section A.11).**
>
> > For instance, I couldn't find which LLM was used,
>
> Sorry for this missing part, GPT4-o was used consistently in our experiments. We added this to the manuscript (pp. 6, line 261).
>
> Thank you again for your insightful feedback, that helped us improve our work. We hope that we clarified your concerns, and we are happy to answer any further questions.

---

> > ### Comment · Reviewer_NFD8 · 2024-11-22
> > **Response to Authors**
> >
> > 1. The mixed method of determining high-level planning/control through logical methods and detail control through RL is very common (just a casual Google Scholar search yields results [1,2,3]). The innovation of this paper lies in replacing the hierarchical structure with a parallel mixed structure, and introducing LLM during logical control (although I doubt how significant the reasoning ability of large models is based on task complexity, the effort of predefined representations, and prompt+example). Therefore, I believe the innovation is not significant.
> > 2. The author also admits that the logical part is responsible for high-level reasoning, while RL is responsible for low-level reactions. The combination of the two is actually time-sharing (meaning that the mix does not bring about new control strategies, but rather just time-sharing control, which is essentially no different from hierarchical mixing and does not yield any conceptual new discoveries). I believe this is also related to the input and action space. I suggest that the author analyze whether the actions output by the logical part and the control part are distributed in different parts of the action space. For example, in Seaquest, the logical part controls ascending and inhaling oxygen, while the RL part is responsible for evasion. This is also related to the logical part receiving macro inputs (object-centered representation) and the RL part receiving image pixel inputs. Therefore, I think perhaps in more complex environments, new discoveries from this mixed method could be obtained?
> > 3. In summary, I partially acknowledge the author's response (perhaps it is not necessary to compare with the strongest RL methods, as it is not very significant, but hierarchical mixed logic and reinforcement learning methods are often experimented in Monte Carlo revenge, which is well-suited for long-term planning + detailed control, so I feel the author's response to complex environments is not correct), but I believe the key concerns have not been addressed. Therefore, I won't increase points just yet.
> > [1] Combining Reinforcement Learning with Symbolic Planning
> > [2] SDRL: Interpretable and Data-Efficient Deep Reinforcement Learning Leveraging Symbolic Planning
> > [3] Symbolic Plans as High-Level Instructions for Reinforcement Learning

---

> ### Author Response · Authors · 2024-11-22
> **Thank you for your response (1/2)**
>
> Dear Reviewer,
>
> Thank you for your prompt answer. We appreciate your attention to detail, as it has helped us identify where our explanations might have caused some confusion. To clarify, **BlendRL does not employ planning within the logic reasoner. It does not build or have access to any world model.** Both the logic and neural components are optimized using the model-free PPO algorithm.
>
> We understand that our reference to System 1 and System 2 might have led you—and potentially future readers—to think that System 1 is optimized by RL while System 2 relies on logic planning. In fact, BlendRL is so far designed to produce model-free RL agents. We have made some changes to the manuscript to better highlight this (p. 2, line 124).
>
> Our statement in the introduction aligns with your observations and the references you suggested: *"The current approach to combining these systems typically uses a top-down (i.e., sequential) method: using deliberative systems (e.g., planners) to provide slow, high-level reasoning to select reactive systems (e.g., Deep RL)"* (cf. lines 58-60). **Our contribution is to move beyond this planner-as-separate-function approach.**
>
> BlendRL ***blends*** logic and neural policies, which we believe is a significant contribution. Our research demonstrates that BlendRL can learn in scenarios where not all necessary priors are available to the agent. When the logic policy isn't sufficient, BlendRL agents rely on their neural components to tackle tasks, as shown in our experiments. We believe that this offers valuable insight for the ICLR community, and to our knowledge, no other paper showcases the use of neural policy to offset suboptimal logic policies.
>
> Thus, we agree with your argument: *”... doubt how significant the reasoning ability of large models is based on task complexity"*, and propose a way to circumvent this problem, using the prior-free neural networks.
>
> To your 3rd point, we indeed do not perform any planning or model-based RL, for which Montezuma's Revenge can indeed be an accurate benchmark.
>
> Let us address each of your concerns.
>
>
>
>
> > The mixed method of determining high-level planning/control through logical methods and detail control through RL is very common (just a casual Google Scholar search yields results [1,2,3]). … Therefore, I believe the innovation is not significant.
>
>
> We appreciate the suggestions. However, these papers aim to integrate pure symbolic planning within RL, while we do not. Thus, these papers’ results do not degrade our claims and evaluations. Abstracts of each paper already demonstrate this:
>
>
> [1] *“The planner shapes the reward function, and thus guides the Q-learner quickly to the optimal policy.”*
>
>
> Namely,  the planner is used to produce reward functions to guide the Q-learner agents. This is a fundamentally different approach from BlendRL. In BlendRL, the (learnable) logic policy produces action distributions directly from observation, just as the neural one does, and both policies are trained jointly via RL signals.
>
>
> [2] *“This framework features a planner – controller – meta-controller architecture, which takes charge of subtask scheduling, data-driven subtask learning, and subtask evaluation, respectively.”*
>
>
> As illustrated in Fig. 1 of [2] (p. 4), the symbolic planner operates as an independent module and is not trained through RL signals. Additionally, the use of PDDL and CLINGO indicates reliance on pure symbolic planning and answer set programming, suggesting the symbolic module functions as a static component. On the contrary, **BlendRL aims to integrate learnable symbolic policies with neural ones and train them jointly.**
>
>
> [3] *“This work explores the use of high-level symbolic action models as a framework for defining final-state goal tasks and automatically producing their corresponding reward functions.”*
>
>
> Namely, the planner synthesizes reward functions that guide hierarchical RL agents. The same argument as in [1] applies here, and this approach is fundamentally different from BlendRL.
>
>
> Overall, these papers use symbolic planners (and logic solvers) to realize efficient training of deep agents in RL (e.g. better sample efficiency) by guiding them via generated plans and subtasks. This approach will pose the bottleneck that BlendRL addressed. For example, the agents must be trained once again when environments are modified (as long as the policy itself is purely neural), as the symbolic components are not directly integrated into a policy. In contrast, BlendRL agents can adapt to modifications (e.g. randomized ladder positions in Kangaroo) immediately without additional training, as we have shown in experiments (Q3, pp.8, lines 368-390), due to its hybrid policy representation.  Integrating planning within BlendRL is an exciting direction. However, it is not the focus of this paper.

---

> ### Author Response · Authors · 2024-11-22
> **Thank you for your response (2/2)**
>
> > The author also admits that the logical part is responsible for high-level reasoning, while RL is responsible for low-level reactions.
>
>
> This is not correct.
> **We claim to establish RL for blended neural and symbolic policies. Symbolic policies are jointly trained via RL signals.** The reviewer’s statement holds for the mentioned planning approach in which the planner is a separate external module, but not for BlendRL.
>
>
> > .. meaning that the mix does not bring about new control strategies, but rather just time-sharing control, which is essentially no different from hierarchical mixing and does not yield any conceptual new discoveries
>
>
> Our blended policy representations empower agents to reason and learn by leveraging the strengths of both neural and symbolic modeling. BlendRL agents can even discern when to utilize each type of policy. This approach fundamentally differs from "hierarchical mixing," which does not allow agents to switch between symbolic and neural modeling on a frame-by-frame basis. Additionally, hierarchical mixing treats symbolic components as static external systems, whereas BlendRL integrates them on the same level as neural policy, allowing joint training through RL signals.
>
>
> > I suggest that the author analyze whether the actions output by the logical part and the control part are distributed in different parts of the action space. For example, in Seaquest, the logical part controls ascending and inhaling oxygen, while the RL part is responsible for evasion.
>
>
> **Once again, both the logic and neural components are optimized through the model-free PPO algorithm, using the reward signal.**
>
>
> **We have already reported the results of this analysis in Fig 6 (pp. 9 lines 376-) on *Seaquest* and *Kangaroo* in our jointly-optimized neural and logic policies.** A demonstration of a working agent is available in [anonymous codebase](https://anonymous.4open.science/r/anon-blendrl-BA06/README.md), *demonstrating what the reviewer suggested: the logic policy takes over to rescue divers while neural policies focus on eliminating sharks*. We observed this complementary nature also on explanations (*cf* [explanation demo](https://anonymous.4open.science/r/anon-blendrl-BA06/explanations/README.md) and our reply to Reviewer pgu6).
>
>
> Moreover, to elaborate, we visualize the action distribution (the proportion of each action) of a trained BlendRL agent in Seaquest (for 10k steps). See [anonymous link](https://anonymous.4open.science/r/anon-blendrl-BA06/assets/seaquest_action_distribution.pdf). We observed that, while the neural policy primarily executes reactive actions (e.g., fire), the logic ones predominantly handle complementary actions. We added this discussion in the revised manuscript (pp. 30, l.1380-1398).
>
>
> > Hierarchical mixed logic and reinforcement learning methods are often experimented in Monte Carlo revenge, which is well-suited for long-term planning + detailed control, so I feel the author's response to complex environments is not correct)..
>
>
> Again, we are not addressing the difficult exploration of Montezuma's Revenge, as tackling long-term planning with sparse reward environments is not our contribution. **BlendRL agents do not incorporate a planner. We utilize environments where distinct skills can be identified. We demonstrate that neural networks enable the BlendRL framework to learn complementary skills that the logic components cannot, through joint learning on RL signals.**
>
> We hope this clears any misunderstading and demonstrates why our approach is different from the approaches mentioned by the reviewer. We will be happy to answer any further questions in the rebuttal phase and hope that the reviewer reconsiders the rating.

---

> > ### Author Response · Authors · 2024-11-25
> > **Any further questions?**
> >
> > Dear Reviewer,
> >
> > Since the discussion phase is coming to a close, we would like to ask if there are any outstanding concerns. We have provided detailed comments to your concerns and hope that we have cleared all misunderstandings.
> >
> > If yes, it will be great if you could reconsider your score.
> >
> > Regards,
> >
> > Authors

---

> ### Comment · Reviewer_NFD8 · 2024-11-25
>
> I appreciate the author's reply, which explains some unclear descriptions in the original paper. I hope the author can make changes in the revision paper. But I still have doubts about the lack of innovation in logic + reinforcement learning and the effectiveness of the method in relatively complex tasks. I am willing to raise the score to 5.

---

### Author Response · Authors · 2024-11-21
**General Remark**

Dear Reviewers,

Thank you for your fruitful and valuable feedback. We sincerely appreciate the time and effort you have devoted to helping us refine our paper. Let us clarify our contribution and provide more comparison to deep and symbolic agents.

## BlendRL's contribution and chosen environments
Our research builds on the well-established logic-based neuro-symbolic RL literature. Notably, NLRL (ICML 2019) [1] and NUDGE (NeurIPS 2023) [2] are pivotal works in integrating differentiable logic into RL agents. These studies have primarily been assessed in synthetic or simple environments (e.g. blocks world), which do not require a solid combination of System 1 and System 2. Despite this limitation, they have been published in top-tier conferences, as their transparent yet learnable policy representations offer significant insights to the community. However, applying these approaches to more complex tasks is challenging and remains a significant bottleneck, mainly due to their reliance on expert-provided inductive bias (*i.e.* concepts such as *closeby*) necessary to solve the tasks. BlendRL agents address this shortcoming in two by:
* (i) utilizing LLMs to obtain these relevant symbolic concepts and their associated logic rules.
* (ii) integrating neural policies that can compensate for a potential lack of these necessary concepts.

The chosen Atari environments challenge the agents' as they incorporate multiple objectives (e.g., shooting enemies, managing oxygen, and collecting divers in *Seaquest* [3]). They allow us to demonstrate that BlendRL agents can meaningful logic policies (as we showcase their rules) *and* even beyond if not provided with all the necessary context. While we are currently extending the set of testing environments (e.g. to *QBert* as suggested by Reviewer pgu6), we believe that the existing results already validate our paper’s claim.

Our contribution focuses on allowing logic-based RL policies to work harmoniously with neural ones, aligning them more closely with Kahneman's System 1 and System 2 concepts [4], and not on developing the SOTA RL algorithm. To this end, BlendRL aims to create transparent and robust agents that combine the strengths of both neural and logic information processing methods.

## More comparison to deep and neuro-symbolic baselines
We agree that comparing BlendRL to additional deep learning and neuro-symbolic baselines could enhance the validation of our proposed method. Consequently, we present comparisons with **6 additional baselines** in the table below. Our evaluation includes 2 deep learning, 5 neuro-symbolic, and 1 human baselines against BlendRL.  We trained NLRL [1] agents in the selected environments. For other baselines, we report the performance as described in their respective papers if available, or extend the evaluation based on their provided code. We are also currently working on obtaining results of INSIGHT agents in *Kangaroo* and *DonkeyKong* [7], that will be included in the camera ready version.
|Environments|DQN[8]    |PPO        |NLRL*[1]|NUDGE[2]|SCoBots[5]  |Interpreter[6]|INSIGHT[7]  |BlendRL  |Human[8]|
|------------|----------|-----------|--------|--------|------------|--------------|------------|---------|---------|
|Kangaroo    |2696      |790.0±280.8|3034±11 |3058±25 |4050±218|1800±0    |-           |**12619**±132| 2739    |
|Seaquest    |2794      |837.3±46.7 |75±0    |64±0    |2411±377|1093±155  |2666±728.2|**4204**±10  |4425    |
|DonkeyKong  |253.3±45.1|2080±1032  |29±0    |122±25  |426.7±64.3  |1838±459  |-           |**3541**±43  |7320[9]    |


*References*

-------

[1] Jiang, Z., & Luo, S. (2019). Neural logic reinforcement learning. ICML.

[2] Delfosse, et al. (2023). Interpretable and explainable logical policies via neurally guided symbolic abstraction. NeurIPS

[3] Bacon, et al. (2017). The oqption-critic architecture. AAAI

[4] Daniel, K. (2017). Thinking, fast and slow.

[5] Delfosse, et al. (2024). Interpretable concept bottlenecks to align reinforcement learning agents. NeurIPS.

[6] Kohler, et al. (2024)  "Interpretable and Editable Programmatic Tree Policies for Reinforcement Learning." Workshop on Interpretable Policies in Reinforcement Learning@ RLC.

[7] Luo, et al. (2024) End-to-End Neuro-Symbolic Reinforcement Learning with Textual Explanations. ICML.

[8] Nair, et al. (2015). Massively parallel methods for deep reinforcement learning. ICML Workshop

-------

Thank you once again for your insightful comments. We have addressed the concerns raised by each reviewer in our responses. The corresponding revisions in the updated manuscript are highlighted in *blue*.

---

### Meta-Review · Area_Chair_e1sF · 2024-12-19

**Metareview:**

This paper presents BlendRL, a framework that integrates neural and symbolic policies for reinforcement learning, demonstrating improved performance over both pure neural networks and symbolic baseline approaches. The majority of reviewers praised the paper's clear presentation, novel integration of neural-symbolic components, and comprehensive empirical validation in Atari game environments. Initial concerns about technical novelty and comparison with existing work were adequately addressed through additional experiments and detailed clarifications of BlendRL's unique hybrid policy learning approach. The authors were highly responsive during the discussion period, providing thorough analysis of the symbolic-neural policy interactions and making significant improvements to the manuscript.

While there is room to expand evaluation to more complex environments in future work, I recommend acceptance based on the paper's strong technical contribution and the authors' comprehensive response to reviewer feedback.

**Additional Comments On Reviewer Discussion:**

The majority of reviewers praised the paper's clear presentation, novel integration of neural-symbolic components, and comprehensive empirical validation in Atari game environments. Initial concerns about technical novelty and comparison with existing work were adequately addressed through additional experiments and detailed clarifications of BlendRL's unique hybrid policy learning approach. The authors were highly responsive during the discussion period, providing thorough analysis of the symbolic-neural policy interactions and making significant improvements to the manuscript.

---

### Decision · Program_Chairs · 2025-01-22

Accept (Spotlight)